# Modified ALNS Algorithm for a Processing Application of Family Tourist Route Planning: A Case Study of Buriram in Thailand

**Narisara Khamsing** [1], **Kantimarn Chindaprasert** [1], **Rapeepan Pitakaso** [2] 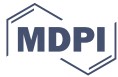, **Worapot Sirirak** [3,*] **and Chalermchat Theeraviriya** [4]

1 Faculty of Tourism and Hotel Management, Mahasarakham University, Maha Sarakham 44000, Thailand; narisara.rmu@gmail.com (N.K.); kantimarn@hotmail.com (K.C.)
2 Department of Industrial Engineering, Faculty of Engineering, Ubon Ratchathani University, Ubon Ratchathani 34190, Thailand; rapeepan.p@ubu.ac.th
3 Department of Industrial Engineering, Faculty of Engineering, Rajamangala University of Technology Lanna Chaing Rai, Chaing Rai 57120, Thailand
4 Department of Industrial Engineering, Faculty of Engineering, Nakhon Phanom University, Nakhon Phanom 48000, Thailand; chalermchat.t@npu.ac.th
* Correspondence: worapotsirirak@rmutl.ac.th

**Abstract:** This research presents a solution to the family tourism route problem by considering daily time windows. To find the best solution for travel routing, the modified adaptive large neighborhood search (MALNS) method, using the four destructions and the four reconstructions approach, is applied here. The solution finding performance of the MALNS method is compared with an exact method running on the Lingo program. As shown by various solutions, the MALNS method can balance travel routing designs, including when many tourist attractions are present in each path. Furthermore, the results of the MALNS method are not significantly different from the results of the exact method for small problem sizes. For medium and large problem sizes, the MALNS method shows a higher performance and a smaller processing time for finding solutions. The values for the average total travel cost and average travel satisfaction rating derived by the MALNS method are approximately 0.18% for a medium problem and 0.05% for a large problem, 0.24% for a medium problem, and 0.21% for a large problem, respectively. The values derived from the exact method are slightly different. Moreover, the MALNS method calculation requires less processing time than the exact method, amounting to approximately 99.95% of the time required for the exact method. In this case study, the MALNS algorithm result shows a suitable balance of satisfaction and number of tourism places in relation to the differences between family members of different ages and genders in terms of satisfaction in tour route planning. The proposed solution methodology presents an effective high-quality solution, suggesting that the MALNS method has the potential to be a great competitive algorithm. According to the empirical results shown here, the MALNS method would be useful for creating route plans for tourism organizations that support travel route selection for family tours in Thailand.

**Keywords:** travel routing design; modified adaptive large neighborhood search; family tourism





## 1. Introduction

At present, the world economy is showing signs of various problems, such as economic recession. The causes of an economic recession include epidemics, natural disasters, and wars. These factors have a significant impact on investment. Tourism is a major target for economic recovery, because it can generate great income due to its low investment and high profitability. Many countries that feature tourist attraction landmarks set important targets for economic revitalization and perpetual development that focus on their tourist landmarks. Tourists from several countries create new travel styles and new experiential

methods of searching. Hotel businesses rely on successful tourists for growth. Tour planning is necessary when group or family trip styles become popular and increase in size or complexity [1]. Tourism informational systems are an essential part of making decisions pertaining to visiting attractive tourist locations, which are also referred to as "landmarks". The ranking, sequencing, routing, and selection of sights to see within a given time window are difficult processes in terms of satisfying the needs of all group members [2].

Many researchers have studied tourism planning and trip routing design within a determined time window for the software generation of tourism planning systems. The software of tour planning design and trip routing design has focused on the balancing of sequences of visits within a time window to support visiting the greatest number of landmarks with the lowest travel distance and expense. This helps travelers to more easily make decisions regarding trip routing and place selection [3–8].

A significant problem in tourism planning design is the satisfaction of all trip members, which is a difficult problem to solve. Therefore, the different tourism styles of family members create a difficult situation in terms of the software creation of a system of tourism planning design. Tourism style is the cause of several factors relating to tour satisfaction. Many factors affect the satisfaction of trip members, such as their different interests, cultures, budgets, time limits, and food and drink preferences [9–11]. Visiting the greatest number of landmarks, low costs, interesting activities, convenient transportation, available facilities, and low travel distances and time costs are important factors for tourism planning design [7,12].

Recently, a heuristic method was applied to solve many tourism problems, which allowed for the generation of a type of tourism planning software that supports a database system of tourism [13,14]. Yu and Chang used a nearest neighbor algorithm to find tour planning solutions for a mobile tourism application. After the solving process, the application could generate an individual trip route, with various suggestions for landmarks and hotels [15]. In 2015, Gavalas et al. employed a slack route tour planning algorithm to create the eCOMPASS system, which was designed for use on a webpage or mobile application in Athens (Greece) or Berlin (Germany). Their algorithm could perform an attractive place selection in urban areas with both efficiency and diversity [16]. The Chinese tourism industry found that service quality reduction and stranded tourists were the main problems in 2019, and better tourism management was urgently required. Forecasting methods with a back propagation (BP) neural network model and fruit fly optimization algorithm (FOA) have been employed to solve these problems. Models using this prediction method display tourism demands accurately for sightseeing spots during high seasons [17].

Tourism has become an important part of the national policy of Thailand due to the continuous growth of the tourism industry there, especially Buriram province in Thailand, which is a popular city. The number of visitors in Buriram increases every year, and most tourists prefer to travel together as a family. The purpose of a family trip is to relax and strengthen family relations; however, family members have various lifestyles, depending on their age and gender. These dissimilarities result in different groups having different family tourism requirements, such as modern technology or a specific tourism style, including the desired spending behavior for products and services. Informational system preparedness is important for allowing family members to search for and select interesting places and accommodation; therefore, travel routes should show information about tourism places and facilities to support decision-making. For this reason, tourism companies in Thailand have created many tourism software applications, with various persuasive purposes and tour styles. Previously, tourism creation has neglected the satisfaction of tourist groups or families regarding tourist sightseeing, which is important for giving tourists a good impression and encouraging them to return. However, many tourism applications cannot meet the requirements for family tours due to the heterogeneity of gender and age in family groups, which brings about different opinions regarding tourism place selection and modes of travel. For the abovementioned reasons, careful tourism routing design for family tours in Buriram is urgently required. Thus, the main contributions of this paper are

twofold. Firstly, we study the family tourism routing problem, according to differences in terms of age, gender, and attraction between family members. Although this these aspects were addressed by Zheng and Liao (2019) [14], the distinctions can be listed as follows: (1) The previous research formulated multi-objective function, the solution provides various alternative routes then user has to select the best route and decision a beneficial trade-off between total utility of the group (TUG) and fairness of individual member (FIN) within a time budget. Current research formulates single objective function and provides the best route in order to make it easy for the user to make decisions. Besides, we include travelling cost in the objective function to economize on budget. (2) We add special constraint to let the tourist indicates the wish list (the place that tourist need to visit) into the route, whereas the literature did not mention. (3) The solution approaches between each research are different. The previous research applied NSACDE which combines ant colony optimization (ACO) and differential evolution algorithm (DEA), whereas we introduce the MALNS as a solution approach. Secondly, we have modified the adaptive large neighborhood search (MALNS) algorithm to solve this problem. Our MALNS algorithm is unique, with modified operators and solution acceptance methods for achieving optimal overall satisfaction regarding tourism places and tourism routing. Finally, the family tourism application base on the MALNS algorithm is flexible in terms of rerouting and re-selecting tourism places in accordance with the changing satisfaction requirements of family members, including in non-family tourism route planning.

This article aims to solve the aforementioned problems by generating a mathematical model and MALNS algorithm for family tours. The MALNS algorithm is carefully route designed and balances the different aspirations of family members and tourism places in routes, depending on differences of age, gender, and time limits. The results from the lingo program using the exact method were compared with the results from MALNS in terms of the performance of the solution-finding method. A mathematical model and modified adaptive large neighborhood search (MALNS) algorithm are applied here to support the creation of software for the processing of a family tourism application. The literature review, mathematical model design, MALNS algorithm, computational consequences and result of the algorithm, conclusions, and recommendations are presented in Sections 2–6, respectively.

## 2. Literature Review

The expansion of the tourism industry is very fast, as it is backed by the government sector. Family tourism is an important policy of tourism organization, as it causes the creation of family bonds, family togetherness, and family memories. The family tourism trend has extended around the world. Recently, Schanzel, and Yeoman [1,18] presented a family tourism incentive, including differences between family members in terms of age, immigration, multi-generational travel, social capital and the creation of memories, helicopter parenting, experiential family holidays, children, blended families, new family markets, and gender. These are factors that influenced the creation of family tourism and family tourism reorganization. In particular, the age and gender of a family member are the primary factors in the determination of his/her complex passions and particular needs. Wu et al. [19] presented a family tourism experience for Chinese children aged between 8–11 years. They found that this group of children had an apparent favorite activity, and several family tourism groups experienced that their tourism experience emphasized physical activities, family togetherness, animal interaction, food testing, and the appreciation of the natural and built environment. These activities influence the generation of destinations that are interesting as family tourist attractions. In addition, Ingkadijaya [20] studied family tourism requirements in relation to travel motivations, family tourism types and activities, and the travel motivations of families in Bogor city. The main factors were the differences in the age and number of family members. The data for analysis were collected using a survey approach, and statistics are used for the description. It was found that two major family travel motivations are family members'

desire to free themselves from their personal lives and develop family relationships. The tourism types and activities are cultural, natural, and special activities associated with family tourism. The travel motivation of a family is relaxation and the development of family relationships through participation in desirable activities. Additionally, Lima et al. [21] explain that family tourism in Portugal is focused on the development of family relationships through participation in new activities, relaxation, and exploration of new environments. An influencing factor that affects the creation of experiences and travel choices in family tourism is the economic difference between families. This factor is an important factor and should not be neglected in designing family tourism styles and travel. This reason helps to reduce the differences in family tourism and raise people's quality of life. From our review of the literature on family tourism, the factors that affect family tourism are differences in terms of gender, age, family activities, spending, destinations that are considered to be attractive, and travel motivations. These factors constitute a determinative factor of member satisfaction in family tourism experience.

However, due to the effect of the generation of experienced complacence in planning family travel experiences on family member expectation, it is difficult to strike a satisfying balance in family tourism. To achieve such a satisfying balance in a family tour, it is necessary to consider the importance of mental factors and relations within the family. Therefore, family tourism design and planning is important and effective in achieving the purpose of adapting to the diverse desires of family tourists. Family tourism design and planning is a tourism problem, which is a very difficult problem to solve due to the diverse desires of family members. In tour design and tour route planning, the difficulty of tourism place selection for sightseeing and family activities must be solved in order to achieve a reasonable level of satisfaction in each family member. Consequently, tourism destination affects tour trip design and tour route planning in family tourism, which is a tourism problem that has not been adequately addressed in the literature.

From the tourism problem literature review conducted here, two primary problems were found for landmark selection. These problems related to the tour trip design problem (TTDP) and the tour route planning problem (TRPP). The TTDP and TRPP are complex problems due to the large amount of data that must be considered for less popular destinations or new landmarks and facilities. In addition, the many factors influencing the problem complexity are differences in terms of time limit, season, traffic, and travel area, which make finding a solution more difficult. For this reason, a local searching method cannot solve these problems, and a heuristic method is therefore the proper solution method. Han et al. [22] successfully applied a particle swarm optimization (PSO) algorithm to the TRPP to find the maximum tourist satisfaction and the minimum travel distance. The determining factors of this design were the tourism cost, tourism attraction place, and tour time. Moreover, their study found that the congestion levels at the tourist attractions and the numbers of tourists influenced tourist satisfaction for sightseeing. Similarly, Xiao et al. [23] studied a neural net buffer algorithm for landmark selection in the context of tour route selection for the purpose of tourist support in Zhengzhou, China. The constraints were interesting spot classifications, the number of tourism locations, and the visiting time limits. The results showed good route selection and fast decision-making abilities for tourist support. Sirirak and Pitakaso [7] presented a study on the optimization of tour distance and the exploration of suitable marketplaces in a determined time window. An ALNS algorithm was used for problem solving in marketplace location selection and route design in Chiang Rai, Thailand. The best solution found by the ALNS algorithm included six approaches for destruction and five approaches for reconstruction. The ALNS algorithm exhibited a 1.12% higher efficiency than the exact method, and the processing time of the ALNS algorithm was about 99% faster than the exact method. Therefore, ALNS algorithms offer high-quality solutions, but the tourism problem solving research on single tourists, as mentioned above, is not applicable to problem solving in family tourism.

Recently, family tourism-related research has focused on solving the TRPP, TTDP, and fashion development continuously in a tour group. Yu et al. [24] displayed the trip selective

problem of tourists in team orienteering using several transportation modes within certain time windows and showed that this problem is an obstacle to tourist trip design. The mixed-integer programming model based on the TOPTW is called the multi-modal team orienteering problem with time windows (MM-TOPTW). The MM-TOPTW was applied in their solution. A two-level particle swarm optimization (2L-PSO) method was used to solve the TTDP with multiple social learning terms, and transportation time and cost were considered as factors. The results showed a higher quality solution when compared to other algorithms. In the same year, a greedy approach was employed for the TRPP, proposing the time and budgets of tourists as the determining factors. Appropriate tourism routes were generated via this approach. Besides, the results showed the influence of the traffic in a tourism route on the tour time [9].

Zhu et al. [25] and Liao et al. [26] presented a solution to the TTDP, considering optimized routes and time allocation, with a genetic algorithm (GA) and differential evolution (DE). The problem they encountered was the variation of the time and the environment, including the travel time or waiting time. Time and environmental changes affect the sightseeing choices of tourists. The studies found a suitably interesting point for routing to support travel and tourist confidence; however, conflicting lifestyles were the main problem for tour groups with different desires. Recently, Zheng and Liao [14] presented a personal tour route design scheme using a mixed method of ant colony optimization (ACO) and a DE algorithm. The solving problem for tour route design was balancing the satisfaction of a heterogeneous tourist group at Kulangsu, an island off the coast of Xiamen in China. Their results show high-quality solutions and more diversified, realistic, and personal routes. Moreover, their research has shown that a primary influencing factor in route design is real-time crowd prediction in popular tourism locations and hotels. This is a factor that affects personalized route design in urban tourism and has not been studied in the context of group tourism problem solving.

Similarly, Hu et al. [27] presented a study tour recommendation using a sequential design of attractive destinations for travel based on tourist group guidance. The two methods of sequential generation of attractive destinations are Attention-based Tour Group Recommendation (AGREE) and bi-directional recurrent unit (Bi-GRU). The result shows the performance of generating an attractive destination sequence on the basis of a real-world dataset. Our review of the tour group literature showed several influencing factors in tour group development, including transportation type, cost, tour time, traffic in the tour route, environment, satisfaction, tourism location, and accommodation. These factors are not applicable to family tourism. In addition, the high-performance diversified heuristic method of tour group problem solving can be divided into single and combination methods, such as PSO, GA-DE, ACO-DE, and AGREE-Bi-GRU. Clearly, combination methods are used to improve the process of finding solutions to the TTDP and TRPP. However, the abovementioned method has a limitation associated with the improvement of solutions, which is that it can only use a certain number of heuristic methods.

Several research suggestions highlight the study of influencing factors and not just the adjustment of problem solving for the TTDP and TRPP alone. The factors include the congestion level of the tourist destination, the conflicting lifestyles of tourists, the traffic on the tour route, real-time crowd prediction, and the hotels in the tour route. Furthermore, differences in satisfaction, age, and gender are worthy of consideration for family tourism, as they are factors that should not be ignored. In addition, the literature review shows that the most effective algorithms for tourism design are heuristic methods. Many studies have shown the high effectiveness of processing in terms of problem solving, showing that it has the best solutions and fastest times. The results of several heuristic methods have been accepted on the basis of their effective solutions. Especially, ALNS algorithms display a prominent point of diversity in terms of destruction and reconstruction methods. Additionally, the ALNS method has a procedure of solution acceptance based on a heuristic approach, which only accepts the best solution. Therefore, the ALNS algorithms generate highly effective solutions, and the required processing time is a distinctive point of problem

solving using the ALNS method. However, the ALNS method has a variety of solution improvement approaches and solution acceptance processes to achieve the best solution, but the accepted solution may profit from further solution development in order to achieve an optimal solution.

Therefore, the performance improvement of the process finding and process of solution acceptance of the ALNS method was considered in this paper. The performance development of process finding based on the ALNS method is called Modified Adaptive Large Neighborhood Search (MALNS) for problem solving in this work. The research focus here is to appropriately determine time windows for family tours in terms of the different demand conditions for different levels of satisfaction, ages, and genders.

## 3. Problem Definitions and Mathematical Model

This section explains the problem characteristics and the formulation of the mathematical model applied to the computation of the family route design problem, as shown in Sections 3.1 and 3.2.

### 3.1. Problem Definitions

The family tourism problem in Thailand affects the lifestyles of families. Individuals in families have different inclinations toward various tour styles, which is an effect of their age. The different influences of age and gender cause disagreements in terms of sightseeing point and recreational area selection. Surveying has found that children favor amusement parks and zoo parks, adolescents favor adventure locations, middle-aged people favor natural places or historic sites, and elders generally prefer peaceful places or temples. Therefore, the age and gender of family members influence their satisfaction level regarding a certain sightseeing point.

This highlights the difficulty in tour travel planning. Family members disagree, which affects their decisions relating to preferred sightseeing points, and, in effect, the unsuitable selection of interesting landmarks is a tour route planning problem. Route planning is effective in terms of finding ideal travel routes, hotels, tour times, tour satisfaction, and traveling costs. A problem pattern of family tourism routing is shown in Figure 1a, showing a non-optimal tourism route. Ideal routes consider the opinions of all family members. The number of tourist locations visited in a route presents an unpleasant balancing problem in terms of the different preferences of family members. The primary objective of route design is eliminating disagreement between family members. Therefore, this research deals with the TRPP which is a variants of the orienteering problem (OP). The target of this kind of problem is to provide a routing tour that maximizes score, satisfaction in this case, within a time constraint.

An ideal route is shown in Figure 1b, showing a problem solving process that involves the maximization of the satisfaction of each family member by achieving the highest number of landmarks in the route, with various tourism offerings. The family tour route is constrained by the consideration of finding the maximum overall satisfaction of family members and the lowest travel cost, while still visiting an adequate number of landmarks. This work presents a family tourism problem solving process for travel route planning of landmark visits, rating different levels of satisfaction of family members in a case study of Buriram Province, Thailand.

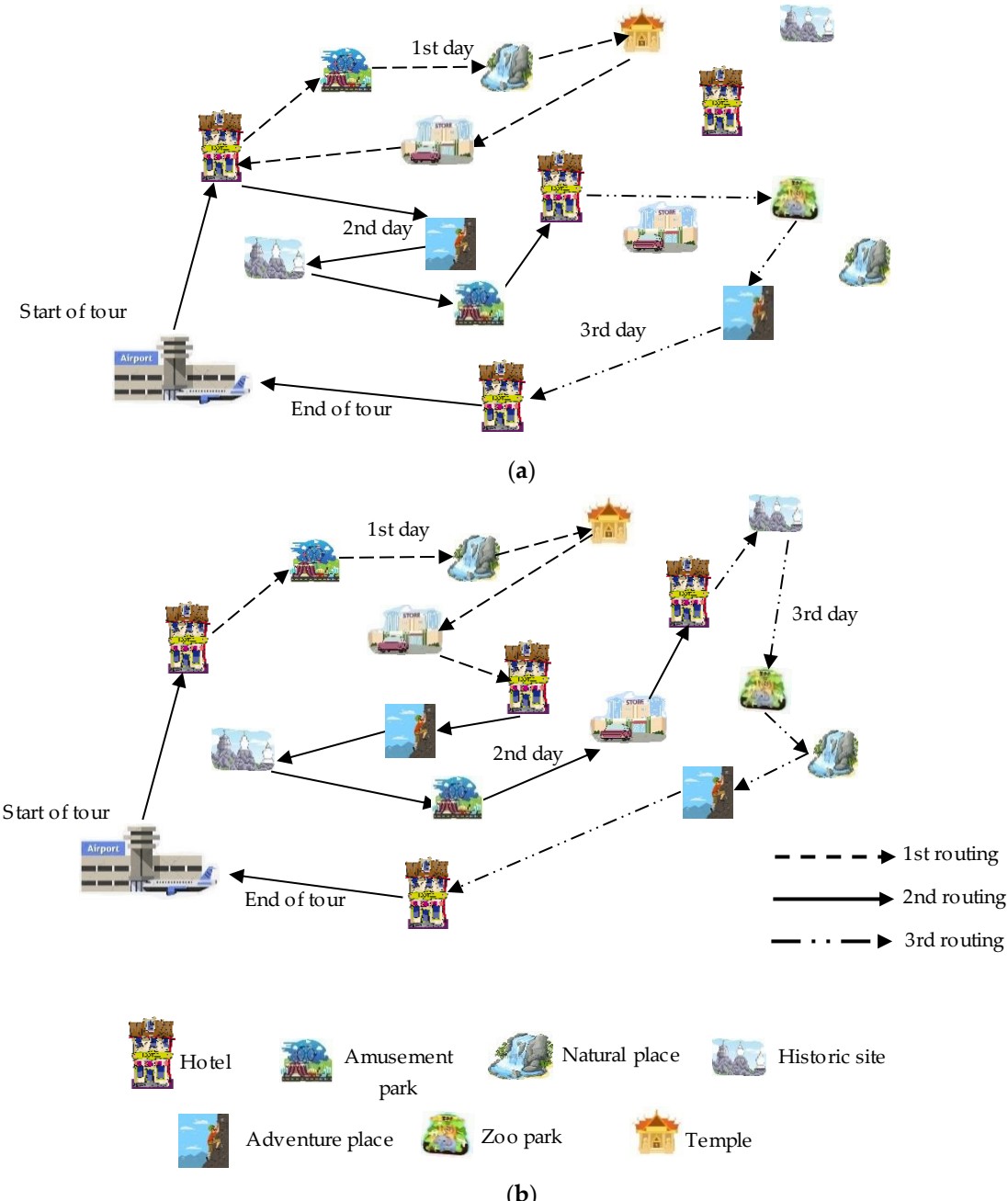

**Figure 1.** Family tourism problem framework (**a**) Problem pattern of family tourism routing. and (**b**) Problem pattern solving.

### 3.2. Mathematical Model

The family tourism routing planning problem is a complex and difficult problem, because there are many tourism attractions with different features that are preferable to different people. A suitable travel route is found via maximizing the satisfaction rating of sightseeing points, while considering the various constraints of family members, and this is an objective of this work. The aforementioned family tour design problems were converted into a mathematical model with the following properties:

Indices

$I$ = set of hotels and travel locations for both entering and exiting, where $i$, $j \in I$;
$K$ = set of tour days, where $k \in K$;
$L$ = set of each tourist, where $l \in L$;

$U$ = set of special search sequences of person categories, such as elders, adolescents, children, and all ages, where $u \in U$;

$V$ = set of search sequences for tourism categories, such as nature, cultural place, adventure place, restaurant, etc., where $v \in V$; $M$ = set of ranges for age and gender, where $m \in M$; $N$ = set of tourism categories, where $n \in N$.

Parameters

$\alpha$ = factor of low-expense travel selection;

$\beta$ = factor of tourism category selection;

$\gamma$ = factor of the selection of the tourism place for different age ranges and genders;

$A_{ik}$ = 1 when the initial point of travel is $i$ on tour day $k$; otherwise, 0;

$B_{ik}$ = 1 when the end point of travel is $i$ on tour day $k$; otherwise, 0;

$C_i^o$ = open time of tourism place $i$;

$C_i^c$ = close time of tourism place $i$;

$D_{ij}$ = distance of travel from node $i$ to $j$ (kilometers);

$T_{ij}$ = travel time from node $i$ to $j$ (minutes);

$S_i$ = sightseeing time of tour place $i$ (minutes);

$R_k$ = sightseeing time of tour day $k$ (minutes);

$E$ = fuel cost of travel via car (baht/kilometer);

$F_{in}$ = 1 when tourism point $i$ is tourism category $n$; otherwise, 0;

$G_{im}$ = satisfaction value of tourism places $i$ for different age ranges and genders $M$;

$Q_{vn}$ = 1 when tourism category $n$ of the search sequences is V; otherwise, 0;

$P_{um}$ = the value of special search $u$ for age ranges and genders $m$;

$W_{lm}$ = 1 when tourist $l$ is within age range and gender $m$ in set $M$; otherwise, 0;

$Y_i^d$ = 1 when the tourism place is in the wish list; otherwise, 0.

Decision Variable

$x_{ijk}$ = 1 if the featured travel is from node $i$ to $j$ of tour day $k$; 0 if otherwise.

$y_i$ = 1 if the featured travel is to tourism place $i$ for a visit; 0 if otherwise.

Support Decision Variable

$t_{ik}$ = time of visit at tourism place $i$ on tour day $k$.

Objective Function

$$Max\ z = \frac{\alpha}{E \sum_{i \in I} \sum_{j \in I} \sum_{k \in K} D_{ij} x_{ijk}} + \beta \sum_{i \in I} \sum_{n \in N} \sum_{v \in V} y_i F_{in} Q_{vn} + \gamma \sum_{i \in I} \sum_{l \in L} \sum_{m \in M} \sum_{u \in U} y_i W_{lm} G_{im} P_{um} \quad (1)$$

where $\frac{\alpha}{E \sum_{i \in I} \sum_{j \in I} \sum_{k \in K} D_{ij} x_{ijk}}$ is a decision factor of distance, $\beta \sum_{i \in I} \sum_{n \in N} \sum_{v \in V} y_i F_{in} Q_{vn}$ is a decision factor for searching with a specified tourism category, and $\gamma \sum_{i \in I} \sum_{l \in L} \sum_{m \in M} \sum_{u \in U} z_{ilm} G_{im} P_{um}$ is a decision factor for special searching, depending on the satisfaction rating levels of different age ranges and genders.

Constraint

$$\sum_{j \in I} \sum_{k \in K} x_{ijk} \leq 1 \ \forall i \in I \quad (2)$$

$$\sum_{j \in I} x_{ijk} = A_{ik} \ \forall i \in I, \forall k \in K \quad (3)$$

$$\sum_{j \in I} x_{jik} = B_{ik} \ \forall i \in I, \forall k \in K \quad (4)$$

$$\sum_{j \in I} x_{ijk} + B_{ik} = \sum_{j \in I} x_{jik} + A_{ik} \ \forall i \in I, \forall k \in K \quad (5)$$

$$t_{ik} - t_{jk} + T_{ij} + S_i \leq R_k \left(1 - x_{ijk}\right) \ \forall i, j \in I, \forall k \in K, i \neq j \quad (6)$$

$$C_i^o \leq t_{ik} \leq C_i^c \; \forall i \in I, \forall k \in K \tag{7}$$

$$\sum_{j \in I} \sum_{k \in K} x_{ijk} = y_i \; \forall i \in I \tag{8}$$

$$y_i \geq Y_i^d \; \forall i \in I \tag{9}$$

$$x_{ijk}, y_i \in \{0,1\} \;\; i, j \in I, \forall k \in K \tag{10}$$

The objective of this model is to maximize the overall family member satisfaction in a tourism place with the Talowest traveling cost, which is detailed below. Objective function (1) is the objective equation for maximum satisfaction, including the low travel value factor for tourism place searching and special searching, depending on the ratings of different age ranges and genders. Constraint (2) pertains to whether each tourism place is visited or not visited. Constraint (3) pertains to the starting point of a travel route on day $k$. Constraint (4) pertains to the return point specification for a travel route on day $k$. Constraint (5) pertains to traveling for visiting and exiting after sightseeing each tourism point $i$ on day $k$. Constraint (6) pertains to the cumulative travel time at $t_{ik}$, which is the cumulative time for travel and visiting and must not be over the prescribed tour time for day $k$. Constraint (7) pertains to the opening and closing times of the tourism locations. Constraint (8) pertains to traveling to the required tourism places. Constraint (9) pertains to wish list places. Constraint (10) pertains to the evaluation of variable decisions.

## 4. Adaptive Large Neighborhood Search Algorithm

Heuristic algorithms consider several problems while finding optimal solutions. An ALNS algorithm has been used to solve the vehicle routing problem (VRP), location routing problem (LRP), and TRPP [7,28–30]. This work considers the family tour route planning problem, also using an ALNS algorithm to find optimal solutions.

### 4.1. ALNS Approach

The ALNS approach here has five steps for finding optimal solutions: Step 1, the first construction is the generation of an initial feasible solution (s); Step 2, the specification of the initial solution (s) is the best solution (s*); Step 3, the operation of destruction and reconstruction is the probability of the selected random initializing weights; and Step 4, the stopping criterion of solution finding is when the new best solution is found. Step 4 includes four sub-steps: Step 4.1, the selection of both destruction (d) and repair (r) is chosen via the weight values for random probabilities; Step 4.2, the new solution is chosen after the destruction (d) and repair (r) of the previous solution (s), where the new solution (s') passes the conformable acceptance condition; Step 4.3, where the new solution (s') becomes the initial current solution(s) for solution finding in the next iteration; and Step 4.4, where the new solution (s') is adjusted with a new weight value when the current solution (s) is better than the former solution (s*). Finally, in Step 5, if the optimal solution is not found, the algorithm returns to the loop in Steps 3 and 4 to find the best new solution until the optimal solution is found. Details of each constituent in the algorithm are shown in Algorithm 1.

An initial solution is constructed with a travel route between a selected place to the next attractive destination, considering the ratings for different age ranges and genders. The initial solution considers the best feasible solution for each route. The algorithm construction for finding a feasible solution is shown in Algorithm 2.

---

**Algorithm 1.** The adaptive large neighborhood search (ALNS) algorithm for family tourism route planning.

---

1.      Construct a feasible solution s;
2.      Set solution s as the best initial solution s*←s;
3.      Assign weights to each operator and perform destruction and reconstruction, with the selective randomization of the initialized weights;
4.      Apply the stopping criterion of solution finding, then

   4.1      Select q, when r ∈ R and d ∈ D, which are related to probabilities p based on the current weights of the applied destruction and reconstruction operators to s;

   4.2      Destroy and repair the previous solution to find a new solution
   $s' = r(d(s))$;

   4.3      If the new solution is accepted by the acceptance criterion, then s←s';
   If solution s is the better than solution s*, then s*←s';

   4.4      Adjust the new weight value of the current solution;
5.      Return to loop 1 to find the new best solution s*.

---

---

**Algorithm 2.** Feasible solution construction.

---

1.      Set $L_t$ <− {1, 2, . . . , n} is comprised of attractive places for travel route construction based on the tourism place ratings.
1.1.    The route construction of travel $r_1$ and determination of the initial solution S ={$r_1$}
2.      Generate $L_t$, which is a non-empty set of routes, for repeated operations.
2.1     Randomly select travel places, where $c_t \in L_t$.
2.2     Insert travel place $c_t$ into $r_k \in S$, where $r_k$ is the best feasible route for the solution.
2.3     If $c_t$ features no feasible inserted place, then create new route S.
2.4     Delete $c_t$ from group set $L_t \leftarrow L_t - \{c_t\}$.
3.      Set $L_f$ <− {1,2, . . . , m}, which is a set of tourism places with good ratings.
4.      While group set $L_f$ is not empty for tourism places:
4.1     Randomly select the tourism place, where $c_f \in L_f$.
4.2     Select tourism place $c_f$ if it is the best feasible location in route $r_k \in S$.
4.3     Delete $c_f$ from the group set $L_f \leftarrow L_f - \{c_f\}$.
5.      Return to loop 1 to achieve the feasible solution S.

---

The construction of a feasible solution is performed in four steps. Step 1 is where $L_t$ is defined as the set of all attractive destinations. Route $r_1$ is generated as an empty route, and the determination of $r_1$ is the initial solution S. Step 2 is where set $L_t$ is used for repeated operations. Step 2 consists of four sub-steps: Step 2.1 is where tourism attractions are randomly selected, forming $L_t$; Step 2.2 is where the destination $c_t$ is inserted into $r_k$, where $r_k$ is the best feasible route for the solution; Step 2.3 is where a new route is created if destination $c_t$ cannot be inserted; and Step 2.4 is where the attractive destination $c_t$ is deleted from $L_t$. Step 3 involves the determination of $L_f$, which is the set of all tourism locations with good ratings. Step 4 involves the set $L_f$, which is the set of all selected tourism locations in a current route. Step 4 involves two sub-steps: Step 4.1 is where tourism place $c_f$ is randomly selected from $L_f$; and Step 4.2 is where tourism place $c_f$ is selected if it is the best feasible tourism place in route $r_k \in S$, with the minimum distance. Finally, in Step 5, if the best feasible solution has not been found, then the process begins again from Step 1.

Finding the best solution depends on the internal operation of each destruction and repair method. Several internal operations for the destruction and repair methods are selected randomly via the cumulative probabilistic weights in the current iteration of the improvement of the solution, as detailed below.

*4.2. Destruction Methods*

The destruction method featured partial solution removal in order to generate new solutions. Here, four approaches for destruction are used. The four approaches were

random removal, worst removal, K-route removal, and relative removal. The differences between these approaches are detailed below.

### 4.2.1. Random Removal

Random removal is an uncomplicated method, where all initial solutions randomly choose a solution for each route, as shown in Figure 2. Step 1 of random removal is where a tourism location is randomly selected for destruction. Step 2 is where the number of locations destroyed is randomly selected. Step 3 is when the random set of points are removed from the route. Finally, Step 4 features the repair method, which is used to find the best solution.

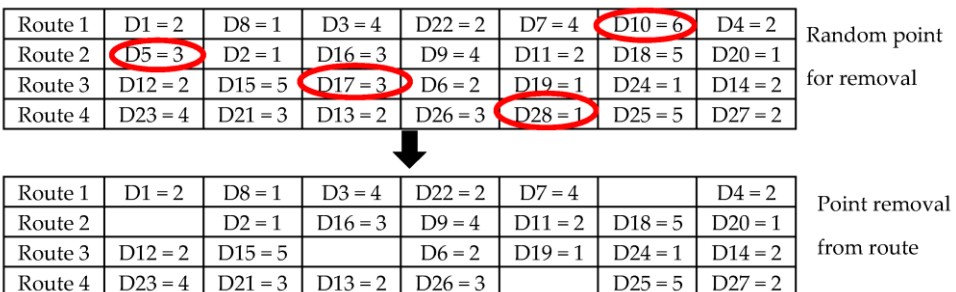

**Figure 2.** Example of random removal for solution destruction.

### 4.2.2. K-Route Removal

The two modes of K-route removal are single-route removal and two-route removal. One of the two modes was randomly selected for the solution destruction, as shown in Figure 3. The K-route removal steps are described as follows: Step 1, a route from all routes is randomly selected. One or more than one route from $R \subseteq S$ is selected, where $|R| = K$. Step 2 represents the determination of the array removal (A) and the array destruction of the randomized route (R). Step 3 is where the removed route is randomized by one of the two modes for solution destruction. Finally, in Step 4, the method proceeds back to Step 1 of route selection to repeat the solution destruction operation and solution finding in set L, which is the solution set for destruction.

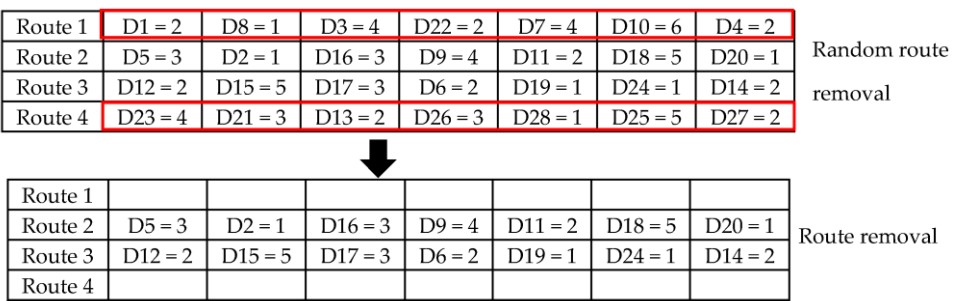

**Figure 3.** Two routes removal example for K-route removal.

The selected route is removed from the set of all routes. After that, the solution repair method generates a new solution. The new random solution is a tourism point in the destroyed route or a point between the destroyed route, where the new solution is inserted into a vacant point in the route.

### 4.2.3. Worst Removal

Worst removal is a solution that destroys the worst solution for each group or route. The priority for removal is determined on the basis of high distances or costs. The worst solution is removed from the route, as shown in Figure 4. The process is repeated as per the process shown in Algorithm 3.

---

**Algorithm 3**. Worst removal algorithm.

---

1. Establish vacancy set L←{};
2. Find set< L, while |L| < q:
2.1 Generate array A, which is an array containing all attractive destinations from *s* not in L;
2.2 Sort A, such that (i < j)→distance (A) [i] < distance (A[j]);
2.3 Randomly select *x* in the interval (0,1);
2.4 Evaluate the addition in L ← L ∪ {A[$x^p$ |A|]};
3. Remove points in L from *s*;
4. Return to loop 1 (set L).

---

| Route 1 | D1 = 2 | D8 = 1 | D3 = 4 | D22 = 2 | D7 = 4 | D10 = 6 | D4 = 2 |
|---|---|---|---|---|---|---|---|
| Route 2 | D5 = 3 | D2 = 1 | D16 = 3 | D9 = 4 | D11 = 2 | D18 = 5 | D20 = 1 |
| Route 3 | D12 = 2 | D15 = 5 | D17 = 3 | D6 = 2 | D19 = 1 | D24 = 1 | D14 = 2 |
| Route 4 | D23 = 4 | D21 = 3 | D13 = 2 | D26 = 3 | D28 = 1 | D25 = 5 | D27 = 2 |

Selection of worst point removal

| Route 1 | D1 = 2 | D8 = 1 | D3 = 4 | D22 = 2 | D7 = 4 | | D4 = 2 |
|---|---|---|---|---|---|---|---|
| Route 2 | D5 = 3 | D2 = 1 | D16 = 3 | D9 = 4 | D11 = 2 | | D20 = 1 |
| Route 3 | D12 = 2 | | D17 = 3 | D6 = 2 | D19 = 1 | D24 = 1 | D14 = 2 |
| Route 4 | D23 = 4 | D21 = 3 | D13 = 2 | D26 = 3 | D28 = 1 | | D27 = 2 |

Worst point removal from route

**Figure 4.** The worst point removal.

Through this process, the worst solution is removed from the route, and the tourism points are then reordered. The worst removal steps are the following: Step 1 is where the first vacant set is established. Step 2 is where set L is found via a repeating process, while |L| < q. Step 2 has four sub-steps: Step 2.1 is where array A is generated, which contains all destinations from the solution that are not in set L; Step 2.2 is where array member A is sorted by distance functional removal; Step 2.3 is where the value of *x* is randomly selected to be between 0 and 1; Step 2.4 is where a tourism point is randomly selected from array A, and the point is evaluated in terms of the set of L, where the value p is a random weight, and low values correspond to increased randomness. Step 3 is where the points in L are removed from the solution. Finally, in Step 4, the value of L is returned to Step 1, where it is used to find further solutions.

### 4.2.4. Relative Removal

Relative removal is the removal of points via relation to the tourism point group. The tourism point in a route is selected by randomization, and the side point is removed as well, as shown in Figure 5. The relative removal algorithm steps are shown in Algorithm 4.

| Route 1 | D1 = 2 | D8 = 1 | D3 = 4 | D22 = 2 | D7 = 4 | D10 = 6 | D4 = 2 |
|---|---|---|---|---|---|---|---|
| Route 2 | D5 = 3 | D2 = 1 | D16 = 3 | D9 = 4 | D11 = 2 | D18 = 5 | D20 = 1 |
| Route 3 | D12 = 2 | D15 = 5 | D17 = 3 | D6 = 2 | D19 = 1 | D24 = 1 | D14 = 2 |
| Route 4 | D23 = 4 | D21 = 3 | D13 = 2 | D26 = 3 | D28 = 1 | D25 = 5 | D27 = 2 |

Selection of Relate point removal

| Route 1 | D1 = 2 | D8 = 1 | D3 = 4 | D22 = 2 | D7 = 4 | D10 = 6 | D4 = 2 |
|---|---|---|---|---|---|---|---|
| Route 2 | D5 = 3 | D2 = 1 | D16 = 3 | D9 = 4 | D11 = 2 | D18 = 5 | D20 = 1 |
| Route 3 | D12 = 2 | D15 = 5 | D17 = 3 | D6 = 2 | D19 = 1 | D24 = 1 | D14 = 2 |
| Route 4 | D23 = 4 | D21= 3 | | | | D25 = 5 | D27 = 2 |

Relate point removal from route

**Figure 5.** Example of relative removal.

Step 1 is where a single tourism point position is randomly selected from all tourism places. The tourism point position is then removed from the set. Step 2 involves the generation of the group set for removal with the initial member $c_t$. Step 3 involves finding L, comprising a process that is repeated while |L| < q. This step has five sub-steps, including: Step 3.1, where one tourism point position is chosen randomly from L; Step 3.2, where array A is created from all solutions but is separate from set L; Step 3.3, where

the members of array A are sorted from lowest to highest by their functional relation R $(c_1, c_2)$. The definable relation is the route distance between $c_1$ to $c_2$ plus the open time of the destination, i.e., $c_1$ and $c_2$ from R $(c_1, c_2) = \alpha$ distance $(c_1, c_2) + \beta \mid t_{ac1} - t_{ac2} \mid$; Step 3.4 is where the value of x is randomly selected to be between 0 and 1; and Step 3.5 is where a tourism point is randomly selected from array A, and the point is increased in set L. The value *p* is a random weight, where low values correspond to increased randomness. Step 4 is where point L is removed from the solution. Finally, in Step 5, the process returns value L to Step 1 in order to find further solutions.

---

**Algorithm 4.** Relative removal algorithm.

---

1. Establish vacancy set L←{};
2. Randomize the centroids p (lat, lng);
3. While $\mid L \mid < q$:
3.1 Find $c_t$ as the approximate position of the centroid point with Euclidean distance;
3.2 $L \leftarrow L \cup \{c_t\}$;

3.3 Centroid update p(lat, lng) = centroids(L) = $\dfrac{\sum_i^{|L|} L[i]}{|L|}$ lat, $\dfrac{\sum_i^{|L|} L[i]}{|L|}$ lng;

3.4 Randomize selection (x) to be between 0 and 1;
3.5 Randomize the selection of the tourism position in array A, and increase the point in set L
4. Remove points in L from solution;
5. Return to loop 1

---

### 4.3. Reconstructive Operations

The four reconstructive operations were used for the insertion of the repairing solution. The four operations were Greedy insertion, Regret-H insertion, Node swap insertion, and Modified node swap insertion. The differences between the methods are detailed below.

### 4.3.1. Greedy Insertion

Greedy insertion is a repair operation that uses the maximum distance or highest travel cost for the purpose of reducing distances or costs to within satisfactory thresholds. Tourism points are reinserted into destroyed routes to repair new routes, as shown in Figure 6.

| Order | Tourism point | | | | | | | Distance |
|-------|------|------|------|------|------|------|------|----------|
| Route 1 | D1 = 2 | D8 = 1 | D3 = 4 | D22 = 2 | D7 = 4 | D10 = 6 | D4 = 2 | 21 |
| Route 2 | D5 = 3 | D2 = 1 | D16 = 3 | D9 = 4 | D11 = 2 | D18 = 5 | D20 = 1 | 19 |
| Route 3 | D12 = 2 | D15 = 5 | D17 = 3 | D6 = 2 | D19 = 1 | D24 = 1 | D14 = 2 | 16 |
| Route 4 | D23 = 4 | D21 = 3 | D13 = 2 | D26 = 3 | D28 = 1 | D25 = 5 | D27 = 2 | 20 |

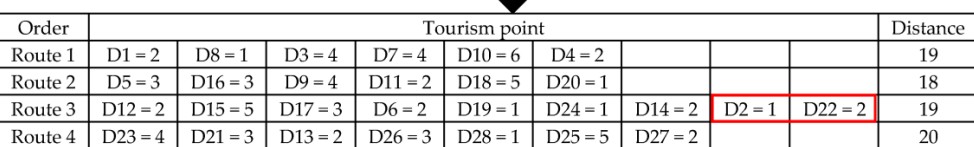

| Order | Tourism point | | | | | | | | Distance |
|-------|------|------|------|------|------|------|------|------|----------|
| Route 1 | D1 = 2 | D8 = 1 | D3 = 4 | D7 = 4 | D10 = 6 | D4 = 2 | | | 19 |
| Route 2 | D5 = 3 | D16 = 3 | D9 = 4 | D11 = 2 | D18 = 5 | D20 = 1 | | | 18 |
| Route 3 | D12 = 2 | D15 = 5 | D17 = 3 | D6 = 2 | D19 = 1 | D24 = 1 | D14 = 2 | D2 = 1 | D22 = 2 | 19 |
| Route 4 | D23 = 4 | D21 = 3 | D13 = 2 | D26 = 3 | D28 = 1 | D25 = 5 | D27 = 2 | | | 20 |

Insertion of the tourism point in the new route.

**Figure 6.** Example of two points of greedy insertion.

The steps of greedy insertion are described as follows: Step 1 involves the determination of value S, which is the solution set of members $\{r_1, r_2, \ldots, r_k\}$. Step 2 involves iterating over each solution ($r_k \in S$), finding the lowest distances and costs with maximum satisfaction for each solution, which are then inserted (c) into $r_k$ $\Delta f_{c,k}$. Finally, in Step 3, the tourism position with the lowest cost or lowest distance of all routes is selected for insertion as per Equation (11):

$$c = \min_{c \in L} \Delta f_{c,k} \tag{11}$$

where c is the lowest travel distance, L is all tourism positions, k is the tourism point number, and f is the distance difference between each route as per the smallest distance evaluation.

### 4.3.2. Regret-H insertion

Regret-H insertion was used here as an improvement of greedy insertion, where $x_{c,h} \in \{1,2,\dots,H\}$ for solution insertion, as per Equation (12):

$$c' = \max_{c\in L}\left\{\sum_{h=1}^{H}\left(\Delta f_{c,x_{c,h}} - \Delta f_{c,x_{c,1}}\right)\right\} \tag{12}$$

where $x_{c,h}$ is the insertion point for tourism point c and the sequential insertion to h. The route exhibits the highest distance with greedy insertion.

The route with the greatest travel distance is selected using the regret-H approach. The greatest distances in the route from tourism point i to j are selected for removal. The solution of regret-H is the inserted position of the lowest distance route, as shown in Figure 7.

| Order | Tourism point | | | | | | | | | Distance |
|---|---|---|---|---|---|---|---|---|---|---|
| Route 1 | D1 = 2 | D8 = 1 | D3 = 4 | D7 = 4 | D10 = 6 | D4 = 2 | | | | 19 |
| Route 2 | D5 = 3 | D16 = 3 | D9 = 4 | D11 = 2 | D18 = 5 | D20 = 1 | | | | 18 |
| Route 3 | D12 = 2 | D15 = 5 | D17 = 3 | D6 = 2 | D19 = 1 | D24 = 1 | D14 = 2 | D2 = 1 | D22 = 2 | 19 |
| Route 4 | D23 = 4 | D21 = 3 | D13 = 2 | D26 = 3 | D28 = 1 | D25 = 5 | D27 = 2 | | | 20 |

            The highest distance point.      The highest distance route

            from the greedy approach.

| Order | Tourism point | | | | | | | | | Distance |
|---|---|---|---|---|---|---|---|---|---|---|
| Route 1 | D1 = 2 | D8 = 1 | D3 = 4 | D7 = 4 | D10 = 6 | D4 = 2 | | | | 19 |
| Route 2 | D5 = 3 | D16 = 3 | D9 = 4 | D11 = 2 | D18 = 5 | D20 = 1 | D25 = 5 | | | 22 |
| Route 3 | D12 = 2 | D15 = 5 | D17 = 3 | D6 = 2 | D19 = 1 | D24 = 1 | D14 = 2 | D2 = 1 | D22 = 2 | 19 |
| Route 4 | D23 = 4 | D21 = 3 | D13 = 2 | D26 = 3 | D28 = 1 | D27 = 2 | | | | 15 |

     Inserted distance point in    The lowest distance in a new route.

     the lowest distance route.

**Figure 7.** Example of Regret-H insertion.

### 4.3.3. Node Swap

Node swap is a local search operator and a heuristic method for local problem solving. Here, two modes of node swapping were used for the solution repair, where point swapping for each route was randomly selected. The tourism points of each route were swapped for the shortest distance or lowest cost, along with the highest satisfaction for new route generation, as shown in Figure 8. The solution with the lowest distance with the highest satisfaction of all solutions for a new route is the best solution.

Random node swap

| Order | Tourism point | | | | | | | Distance |
|---|---|---|---|---|---|---|---|---|
| Route 1 | D1 = 2 | D8 = 1 | D3 = 4 | D22 = 2 | D7 = 4 | D10 = 6 | D4 = 2 | 21 |
| Route 2 | D5 = 3 | D2 = 1 | D16 = 3 | D9 = 4 | D11 = 2 | D18 = 5 | D20 = 1 | 19 |
| Route 3 | D12 = 2 | D15 = 5 | D17 = 3 | D6 = 2 | D19 = 1 | D24 = 1 | D14 = 2 | 16 |
| Route 4 | D23 = 4 | D21 = 3 | D13 = 2 | D26 = 3 | D28 = 1 | D25 = 5 | D27 = 2 | 20 |

| Order | Tourism point | | | | | | | Distance |
|---|---|---|---|---|---|---|---|---|
| Route 1 | D1 = 2 | D8 = 1 | D3 = 4 | D22 = 2 | D7 = 4 | D10 = 6 | D4 = 2 | 21 |
| Route 2 | D14 = 2 | D2 = 1 | D16 = 3 | D9 = 4 | D11 = 2 | D18 = 5 | D20 = 1 | 17 |
| Route 3 | D12 = 2 | D15 = 5 | D17 = 3 | D6 = 2 | D19 = 1 | D24 = 1 | D5 = 3 | 17 |
| Route 4 | D23 = 4 | D21 = 3 | D13 = 2 | D26 = 3 | D28 = 1 | D25 = 5 | D27 = 2 | 20 |

**Figure 8.** Example of node swap insertion.

#### 4.3.4. Modified Node Swap

Modified node swap is an applied approach based on three approaches, including node swapping, cluster removal, and worst removal for solution repair. Firstly, a swap point was randomly selected to be swapped with a similar point. Secondly, the cluster around the selective point was determined. Then, the worst point of each cluster was exchanged among the selected points and the worst point to create a new route. Finally, the worst points of each cluster were swapped with each other, as shown in Figure 9.

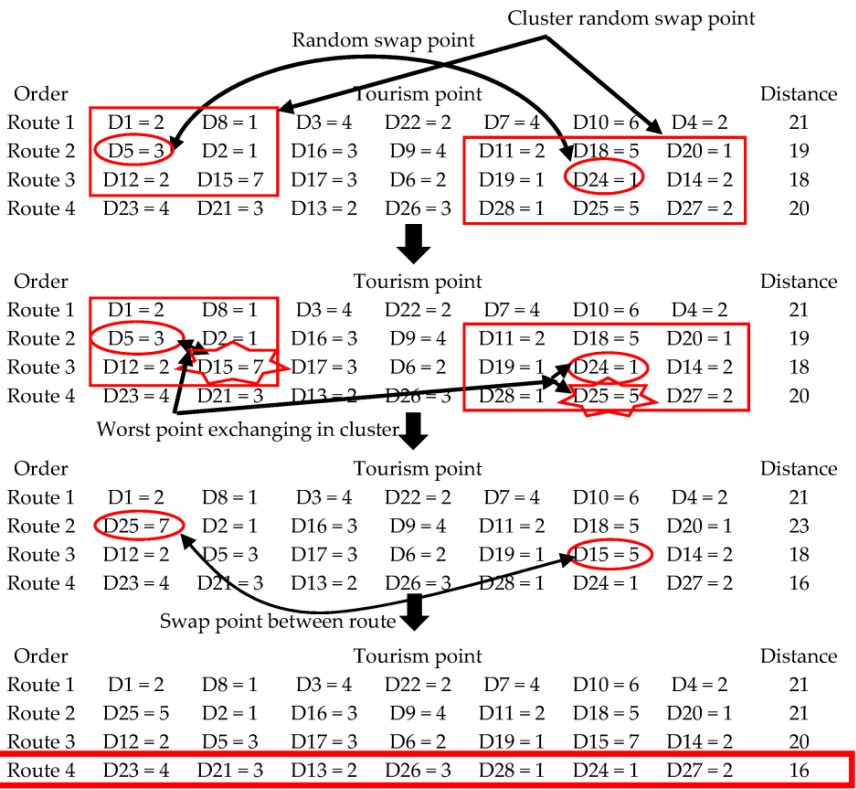

**Figure 9.** Example of modified node swap insertion.

The new route generated on the basis of the lowest distance with the highest satisfaction is the best solution.

#### 4.4. Acceptance Criterion

An acceptance criterion was used to check if each new solution was the best solution. If a new solution was worse than the previous solution, then the acceptance criterion was considered to make a decision as to whether the new solution should be accepted. In this work, the criterion of solution acceptance used three criteria, including the Simulated Annealing (SA) method, Linear Function (LF), and Exponential Function of Current Number of Interactions (EFCNI). One of the three accepted criteria were selected by randomization in each iteration of the solution finding process.

#### 4.4.1. Simulated Annealing Method (SA)

The new solution was checked for acceptance by SA at every iteration [31]. The acceptance of a new solution means that the new solution is better than the current solution. The simulated annealing method for the acceptance criterion considers the probability of new solution acceptance as per Equation (13):

$$P = \exp^{-\frac{(S(V')-S(V))}{T \times K}} \tag{13}$$

where *S(V)* is the best current solution, and *S(V′)* is the newly created solution. The predetermined parameters are *T* and *K*. The new solution is accepted via the probability of the best solution in each iteration of the solution finding process. The acceptance process stops checking solutions when the algorithm does not find a better solution.

### 4.4.2. Linear Function (LF)

The current interaction displays an important matter in accepting or rejection a new solution. The algorithm will search a full area of the search area, and it should have a way to escape from the local optimal when it is trapped in the searching area. Therefore, the data on the interactive current number have been increased by Equation (13). The LF approach shows Equation (14) is as follows:

$$P = 1 - \exp^{-[(\frac{S(V)-S(V')}{S(V)})^2 + (IT - \frac{MaxIT}{2})^2]} \tag{14}$$

where *MaxIT* is the maximum number of iterations, which is the number of current iterations.

### 4.4.3. Exponential Function of Current Number of Interactions (EFCNI)

The EFCNI approach involved checking new solutions for acceptance, which will modify the performance of the algorithm. The exponential function of the current number of interactions is shown in Equation (15):

$$P = 1 - \exp^{-[(IT - \frac{MaxIT}{2})^2]} \tag{15}$$

## 5. Computational Framework and Results

The mathematical model was tested via an exact method using the Lingo program to verify the model accuracy. The exact method is a method for calculating every alternative in solution finding and selecting the best value from every alternative, which is an optimal solution in problem solving. An effective result found via the exact method was compared with the result of the ALNS algorithm. The mathematical model and the ALNS algorithm show the dataset of three primary parameters for testing in Table 1. The determination of the three parameters was carried out via surveying tourists in Thailand. The rating score of interesting tourism place satisfaction was determined as a score between 1–10. The most interesting tourism attractions display high rating scores, and the less interesting tourism attractions show low rating scores. The locations of intermediate interest are represented by a medium score. The second parameter was the number of tourism days, which was between 1–5 days for testing in each experiment, and the number is the precise tourism days fixed by the tourist.

**Table 1.** Dataset input for testing.

| Order | Data Input | Value |
|:-----:|:----------:|:-----:|
| 1 | Rating of tourism place satisfaction | 1–10 |
| 2 | Number of tourism days | 1–5 |
| 3 | Number of family members | 1–5 |

The number of family members used (2–5 persons) was used to determine the age ranges and genders, which were used for testing the mathematical model and the ALNS algorithm. The data for tourism locations and accommodation were selected randomly in the testing of the mathematical model and the ALNS algorithm, as shown in Appendix A Table A1. The result of both the Lingo program and ALNS algorithm were analyzed using the Windows operating system and a computer with an Intel<sup>(R)</sup> Core i5-9500 processor CPU operating at 3.00 GHz with 8 GB of RAM. Each problem size featured similar input parameters for comparing the results of Lingo and those of ALNS. The experiment was

separated into three case problems, including a small problem size, medium problem size, and large problem size. For this work, the definition of each problem size depended on the number of tourism places, which was divided into three ranges: 1–39, 40–79, and 80, which are the small problem size, medium problem size, and large problem size, respectively.

### 5.1. Comparison Method

The results for the three problem sizes were evaluated in terms of their effectiveness in finding solutions. The three sets of problem sizes, small, medium, and large, represented 10, 40, and 80 locations, respectively. Each problem size was tested with 5 instances. A comparison of the results found that the MALNS algorithm effectively found solutions for all problem sizes. In particular, both medium and large problem sizes showed a faster processing time with the MALNS algorithm than the Lingo program. The comparison results for all problem sizes are shown in Table 2. The comparative results for the MALNS algorithm and Lingo program show the global optimal solutions for both methods for small problems. The solutions display evaluations of routes with the lowest total cost of travel and the maximum total satisfaction rating for travel. In addition, the processing times were similarly fast, displaying no differences between the solutions of both methods because of the low number of tourism places when considering small problem sizes. The results for both the total travel cost and total satisfaction rating for travel exhibit slight differences between the MALNS algorithm and the Lingo program for medium problem sizes, where the Lingo program showed longer processing times than the MALNS algorithm. This is due to the increased number of tourism places considered, representing an extended, complex, and difficult problem for the Lingo program in terms of finding a global solution. The Lingo program showed an average processing time of 31.42 h, while the MALNS algorithm showed an average processing time of 1.08 min. Therefore, the fast effective processing of the MALNS algorithm is the most suitable route planning when the number of tourism places is increased and, with it, the complexity of the travel route.

**Table 2.** The comparison results for the Lingo program and the modified adaptive large neighborhood search (MALNS) algorithm.

| Problem Sizes | No. | Parameters | | | Upper Bound, Generated by Lingo Program | | | | MALNS | | |
|---|---|---|---|---|---|---|---|---|---|---|---|
| | | Number of Places | Number of Members | Number of Tourism Days | Total Travel Cost (Baht) | Total Rating of Travel Satisfaction | Processing Time (h) | Status | Total Travel Cost (Baht) | Total Rating of Travel Satisfaction | Processing Time (h) |
| Small | 1 | 10 | 3 | 1 | 450.80 | 10.18 | 00:00:01 | Global opt. | 450.80 | 10.18 | 00:00:04 |
| | 2 | 10 | 4 | 1 | 640.80 | 14.46 | 00:00:02 | Global opt. | 640.80 | 14.46 | 00:00:12 |
| | 3 | 10 | 3 | 1 | 788.40 | 11.30 | 00:00:01 | Global opt. | 788.40 | 11.30 | 00:00:08 |
| | 4 | 10 | 5 | 1 | 43.20 | 6.80 | 00:00:03 | Global opt. | 43.20 | 6.80 | 00:00:16 |
| | 5 | 10 | 3 | 1 | 622.80 | 10.63 | 00:00:06 | Global opt. | 622.80 | 10.63 | 00:00:20 |
| | **Average** | | | | **509.2** | **10.67** | **00:00:03** | **-** | **509.2** | **10.67** | **00:00:12** |
| Medium | 1 | 40 | 3 | 2 | 1093.46 | 23.28 | 24:39:28 | Feasible | 1095.80 | 23.21 | 00:01:12 |
| | 2 | 40 | 4 | 2 | 1283.11 | 17.98 | 23:27:21 | Feasible | 1284.10 | 17.91 | 00:01:15 |
| | 3 | 40 | 3 | 2 | 709.78 | 22.60 | 30:49:13 | Feasible | 710.60 | 22.57 | 00:01:08 |
| | 4 | 40 | 5 | 2 | 957.81 | 16.85 | 28:46:29 | Feasible | 959.10 | 16.8 | 00:01:04 |
| | 5 | 40 | 3 | 2 | 426.12 | 21.99 | 50:49:25 | Feasible | 429.00 | 21.96 | 00:01:03 |
| | **Average** | | | | **894.05** | **20.54** | **31:42:23** | **-** | **895.72** | **20.49** | **00:01:08** |
| Large | 1 | 80 | 3 | 3 | 1099.87 | 30.95 | >72 | Upper bound | 1097.80 | 30.89 | 00:02:11 |
| | 2 | 80 | 4 | 3 | 1291.15 | 36.14 | >72 | Upper bound | 1289.10 | 36.12 | 00:02:01 |
| | 3 | 80 | 3 | 3 | 728.16 | 26.49 | >72 | Upper bound | 727.60 | 26.38 | 00:02:05 |
| | 4 | 80 | 5 | 3 | 925.21 | 33.67 | >72 | Upper bound | 929.10 | 33.59 | 00:02:02 |
| | 5 | 80 | 3 | 3 | 410.54 | 25.34 | >72 | Upper bound | 409.00 | 25.29 | 00:02:09 |
| | **Average** | | | | **890.58** | **30.51** | **>72** | **-** | **890.52** | **30.45** | **00:02:05** |

For large problem sizes, the Lingo program could not find a global solution in an adequate amount of time, where the expected time for processing was more than 72 h

because of the large amount of data to be considered. The Lingo program showed upper bounds for both the total travel cost and total travel satisfaction rating. The MALNS algorithm was very effective and fast for finding solutions, with a processing time of 2.05 min. While the results for the MALNS algorithm did not find a global optimal solution, the response solution was an acceptable solution. The Lingo and MALNS algorithm results are further compared in Table 3. The differences between the methods in terms of effectiveness were calculated by Equation (16):

$$\%diff = \frac{AVGresult_{MALNS} - AVGresult_{Lingo}}{AVGresult_{Lingo}} \tag{16}$$

**Table 3.** Comparison of the results for the Lingo program and MALNS algorithm.

| Results | Problem Sizes | Methods | | Method Difference (%) |
|---|---|---|---|---|
| | | **Lingo Program** | **MALNS** | |
| Average total travel cost (baht) | Small | 509.2 | 509.2 | 0 |
| | Medium | 894.05 | 895.72 | −0.18 |
| | Large | 890.58 | 890..52 | −0.05 |
| Average total satisfaction rating for travel | Small | 10.67 | 10.67 | 0 |
| | Medium | 20.54 | 20.49 | 0.24 |
| | Large | 30.51 | 30.40 | 0.21 |
| **Average differential gap of method** | | | | **0.11** |
| Average processing time (h) | Small | 00:00:03 | 00:00:12 | 3 |
| | Medium | 31:42:23 | 00:01:08 | −99.94 |
| | Large | 72:00:00 | 00:02:05 | −99.95 |

The different percentages of the two methods for all problem sizes show less different percentages. The total average travel cost percentage difference was zero or otherwise insignificant for small problems.

The MALNS algorithm showed a slightly inferior result of 0.18% for medium problems and 0.05% for large problems. The percentage difference for the total average travel satisfaction rating was also zero or insignificant for small problems. A slightly inferior result of 0.24% achieved for medium problems, and a result of 0.21% was obtained for large problems. Therefore, the percentage of the average differential gap of the method is 0.11% in Lingo program and MALNS comparison.

For the MALNS algorithm, the processing time percentage difference was 3% for small problems. Medium and large problems showed dramatic decreases in the processing time percentage differences of 99.94% and 99.95%, respectively. Therefore, the MALNS algorithm exhibits a good solving ability for complex problems, finding effective solutions with a fast processing time. The results in Table 2, showing the total travel cost, total satisfaction rating for travel, and processing time for all problem sizes, were analyzed by paired t-tests to evaluate the performance of the two methods.

The paired *t*-test approach considered 0.05 as the level of significance. The statistical results are shown in Table 4 and display results with *p*-values with significance levels over 0.05 for the total travel cost and total satisfaction rating for travel for all problem sizes. For the processing time, the *p*-values for small problems were greater than 0.05, and the two methods were not significantly different in terms of their processing times. The processing times for medium and large problems show *p*-values lower than 0.05, meaning that the processing times for the Lingo program and MALNS algorithm were significantly different. Therefore, the MALNS algorithm exhibits a strikingly effective solution finding process, and the result is not different from an exact method based on the Lingo program.

**Table 4.** Statistical *p*-value results found by the paired *t*-test method when using the results presented in Table 2.

| | *p*-Value | | |
|---|---|---|---|
| **Problem Sizes** | **Total Travel Cost** | **Total Satisfaction Rating for Travel** | **Processing Time** |
| Small problem | 1.000 | 1.000 | 1.000 |
| Medium problem | 0.093 | 0.282 | 0.000 * |
| Large problem | 0.105 | 0.056 | 0.000 * |

Note that * indicates a significant difference.

Furthermore, the differential percentage of each instance shows an extremely weak percentage of the gap error of the total travel cost value. The MALNS algorithm shows high performance in tourism planning problem-solving. However, the MALNS algorithm was compared with the LNS algorithm and the ALNS algorithm for the efficacy testing. Destruction and a reconstruction approach mean the Random removal and the Greedy insertion, respectively, were employed in the LNS algorithm of the problem solving-solution for the best solution finding. Each round of the best solution in the LNS algorithm has no acceptant criterion.

For, the ALNS algorithm randomized one approach from three destructions and three reconstructions for each round the best solution finding. The three destructions are Random removal, Worst removal, and relative removal. The three reconstructions are Greedy insertion, Regret-H insertion, and Node swap. In ALNS process, The SA approach was used as the acceptant criterion for the best solution.

The result of the MALNS algorithm was compared with the result of LNS algorithm and ALNS algorithm in this problem solving, as shown Table 5. From the solution finding, found that the MALNS algorithm display higher performance than LNS and ALNS algorithm for all problem size. The MALNS algorithm give the lowest average total travel cost in all problem size as shown in Figure 10a, and the highest average total rating of travel satisfaction as displayed in Figure 10b.

**Table 5.** Comparison of the differential gaps of the algorithms in tourism planning problem solving.

| Problem | | Algorithms | | | | | | % Differential Gap of Algorithm | | | |
|---|---|---|---|---|---|---|---|---|---|---|---|
| | | LNS | | ALNS | | MALNS | | LNS: MALNS | | ALNS: MALNS | |
| **Sizes** | **Instances** | Total Travel Cost (Baht) | Total Rating of Travel Satisfaction | Total Travel Cost (Baht) | Total Rating of Travel Satisfaction | Total Travel Cost (Baht) | Total Rating of Travel Satisfaction | Total Travel Cost | Total Rating of Travel Satisfaction | Total Travel Cost | Total Rating of Travel Satisfaction |
| Small | 1 | 479.54 | 9.59 | 456.03 | 10.03 | 450.8 | 10.18 | 5.99 | −6.15 | 1.15 | −1.50 |
| | 2 | 662.98 | 13.21 | 647.98 | 14.26 | 640.8 | 14.46 | 3.35 | −9.46 | 1.11 | −1.40 |
| | 3 | 809.87 | 10.87 | 796.99 | 11.15 | 788.4 | 11.3 | 2.65 | −3.96 | 1.08 | −1.35 |
| | 4 | 45.25 | 6.38 | 44.13 | 6.68 | 43.20 | 6.80 | 4.53 | −6.58 | 2.11 | −1.80 |
| | 5 | 653.84 | 10.42 | 629.28 | 10.37 | 622.8 | 10.63 | 4.75 | −2.02 | 1.03 | −2.51 |
| **Average** | | **530.30** | **10.09** | **514.88** | **10.50** | **509.20** | **10.67** | **3.98** | **−5.75** | **1.10** | **−1.68** |
| Medium | 1 | 1149.52 | 22.5 | 1119.98 | 22.97 | 1097.8 | 23.64 | 4.50 | −5.07 | 1.98 | −2.92 |
| | 2 | 1326.71 | 16.71 | 1298.64 | 16.9 | 1285.1 | 17.27 | 3.14 | −3.35 | 1.04 | −2.19 |
| | 3 | 783.29 | 21.97 | 723.12 | 22.05 | 712.60 | 22.3 | 9.02 | −1.50 | 1.45 | −1.13 |
| | 4 | 1061.25 | 15.49 | 968.96 | 15.88 | 959.1 | 16.71 | 9.63 | −7.88 | 1.02 | −5.23 |
| | 5 | 473.65 | 20.12 | 436.78 | 21.13 | 429.00 | 21.71 | 9.43 | −7.90 | 1.78 | −2.74 |
| **Average** | | **958.88** | **19.36** | **909.50** | **19.79** | **896.72** | **20.33** | **6.48** | **−5.00** | **1.40** | **−2.73** |
| Large | 1 | 1199.98 | 30.96 | 1118.02 | 31.13 | 1097.80 | 31.29 | 8.52 | −1.07 | 1.81 | −0.51 |
| | 2 | 1390.26 | 33.75 | 1319.87 | 36.77 | 1289.10 | 37.04 | 7.28 | −9.75 | 2.33 | −0.73 |
| | 3 | 758.10 | 26.47 | 737.91 | 27.99 | 727.60 | 28.40 | 4.02 | −7.29 | 1.40 | −1.46 |
| | 4 | 993.78 | 32.02 | 946.29 | 33.34 | 929.10 | 33.78 | 6.51 | −5.50 | 1.82 | −1.32 |
| | 5 | 452.67 | 24.33 | 415.28 | 25.98 | 409.00 | 26.41 | 9.65 | −8.55 | 1.51 | −1.66 |
| **Average** | | **958.96** | **29.51** | **907.47** | **31.04** | **890.52** | **31.38** | **7.14** | **−6.36** | **1.87** | **−1.10** |

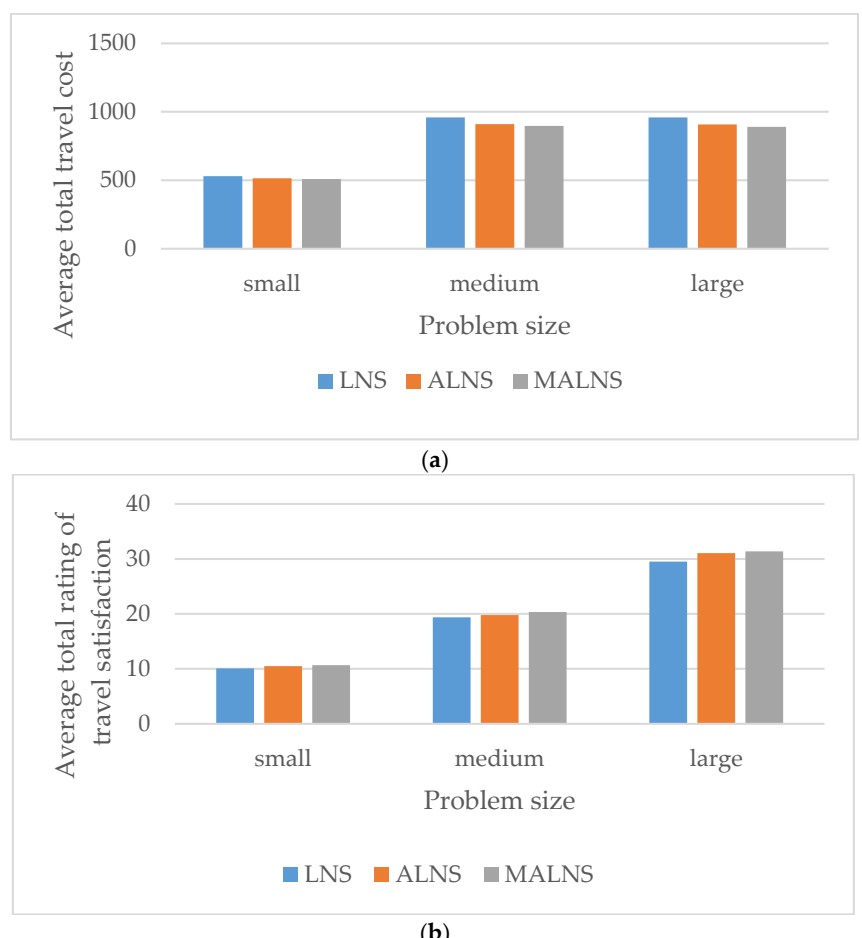

(a)

(b)

**Figure 10.** Tentative comparison of the solution of the algorithms (**a**)Average total travel cost comparison of each problem size. and (**b**) Average total rating of travel satisfaction of each problem size.

For comparison of the average differential gap percentage, the MALNS algorithm shows higher efficacy than the LNS algorithm in all problem size. In the average differential gap percentage, the MALNS algorithm display lower total travel cost than LNS algorithm around 3.98% for small problem size, 6.48% for medium problem size and 7.14% for large problem size. The total rating of travel satisfaction shows the 5.75% average differential gap percentage of small problem size, 5.00% of medium problem size and 6.36% large problem size. MALNS algorithm gives better valuation than LNS algorithm. In comparison of the average differential gap percentage, MALNS algorithm exhibit higher performance than the ALNS algorithm in all problem size. The average differential gap percentage of MALNS algorithm gives lower total travel cost than ALNS algorithm about 1.10% for small problem size, 1.40% for medium problem size and 1.87% for large problem size. The MALNS algorithm also gives the higher total rating of travel satisfaction than ALNS algorithm about 1.68% for small problem size, 2.73% for medium problem size and 1.10% for large problem size.

Therefore, the MALNS algorithm applies the new approach of reconstructive operation and increases the acceptance criterion in the solution finding process, which is a highly effective expansion of optimal solution finding. In addition, the MALNS algorithm exhibits a high performance, compared with the other algorithm, as shown in Table 6, which shows some sample algorithms used in tour group problem solving.

**Table 6.** Comparison of the differential gap percentage of other algorithms used in tour group problem.

| Problem | Approaches | Algorithms | | | | | | | Differential Gap (%) |
|---|---|---|---|---|---|---|---|---|---|
| | | GA | DE | TS | AGREE | ACO | SVD | MALNS | |
| Family tourism planning | Single | | | | | | | ✔ | 0.11 |
| Tour group planning [14] | Combine | | ✔ | | | ✔ | | | 0.49 |
| Tourist group trip design [26] | Combine | ✔ | ✔ | | | | | | 1.98 |
| Design of tour group recommendation [27] | Single | | | | ✔ | | | | 1.30 |
| Tour route design for tourist group [32] | Combine | ✔ | ✔ | | | | | | 3.50 |
| Trip planning for tourist group [33] | Single | | | ✔ | | | | | 0.73 |
| Tour recommendation for group [34] | Single | | | | | ✔ | | | 0.96 |
| Tour group recommendation [35] | Single | | | | | | ✔ | | 0.20 |

Note: GA = Genetic Algorithm; DE = Differential Evolution; TS = Tabu Search; AGREE = Attention-based Tour Group Recommendation; ACO = ant colony optimization; SVD = Singular Value Decomposition; MALNS = Modified Adaptive Large Neighborhood Search.

*5.2. Case Study Area*

In the case study, two categories of tested case study, considering family tourism and non-family tourism route planning problems, were solved using the MALNS method for different result considerations. The similar data inputs for the two types of testing included 2 fixed points in a start-to-end trip, 88 tourism places, 29 hotels, 5 members, 1–10 rating scores for tourism place satisfaction identification, and the fixed day of touring was 3 days. The dissimilar tested data inputs are the factors of the genders and ages of family members, and the total number of family members was determined to be five. The factors of gender and ages are not used for non-families, as shown in Table 7.

**Table 7.** Age ranges and genders.

| Age Ranges | Family | | Non-Family | |
|---|---|---|---|---|
| | Genders | | Genders | |
| | Male | Male | Female | Female |
| 1–24 | 0 | 0 | 0 | 0 |
| 25–35 | 1 | 1 | 0 | 0 |
| 36–45 | 1 | 1 | 0 | 0 |
| 42–60 | 0 | 0 | 0 | 0 |
| >60 | 1 | 1 | 0 | 0 |

This is because the number of tourists of non-family tourism is fixed to zero person. The two categories of data were analyzed using the MALNS algorithm, which has the objective of maximum family member satisfaction and the lowest travel cost. The route planning result of two categories of case studies is shown in Table 8. The family tourism route design for achieving the highest total satisfaction rating included three travel routes. These routes exhibited the highest total ratings for total travel satisfaction, with scores of 44.57. Each travel route displays four positions for points of interest in each day. The first-day tour begins at point E1, which is an airport. The tourism points for visiting are T17, T21, T51, and T40. The point of the end trip is H7, which is accommodation. The satisfaction per person of the first route shows the highest satisfaction rating at 5.16 score of family member age range at 1–24 years because this route specifies this family member age range as the priority. For other family member age range show no different satisfaction score of the family member in the travel route. The second day of the tour travel starts at point H7, and the tourism places are T34, T2, T4, and T16. The H10 point is the point of the end tour, which is the accommodation on the second day. The satisfaction per person of the second route displays the highest satisfaction rating at 4.24 score of family member age

range at >60 years due to this second route set the family member age range at >60 years as the priority of tourism. The final tour day commences at the H10 point of the tour and continues through the T23, T20, T11, and T3 visiting points. The tour ends on the final day at the E2 point, which is a bus station. An individual satisfaction of the final route exhibits high satisfaction rating at 3.87 and 3.69 of family member age range at 36–45 years and 46–59 years respectively due to this route define both family member age range as the priority of tourism. The travel routes featured the minimum travel cost and distance, which were 823 baht and 486 km, respectively. The pattern of travel for the three routes is shown in Figure 11a.

**Table 8.** Case study results for family tourism and non-family tourism.

| Tour Types | Tour Day Order | Route | Start Trip | Tourism Attractions | End Trip | Distances (km) | Travel Costs (Baht) | Rating of Satisfaction | | |
|---|---|---|---|---|---|---|---|---|---|---|
| | | | | | | | | Member Age Ranges | Satisfied Per Person | Total Satisfied Per Route |
| Family | 1 | 1 | E1 | T17, T21, T51, T40 | H7 | 156 | 264 | 1–24 | 5.16 | 15.18 |
| | | | | | | | | 25–35 | 3.32 | |
| | | | | | | | | 36–45 | 2.13 | |
| | | | | | | | | 46–60 | 2.21 | |
| | | | | | | | | >60 | 2.36 | |
| | 2 | 2 | H7 | T34, T2, T4, T16 | H10 | 147 | 249 | 1–24 | 2.34 | 14.73 |
| | | | | | | | | 25–35 | 2.33 | |
| | | | | | | | | 36–45 | 2.56 | |
| | | | | | | | | 46–60 | 3.26 | |
| | | | | | | | | >60 | 4.24 | |
| | 3 | 3 | H10 | T23, T20, T11, T3 | E2 | 183 | 310 | 1–24 | 2.28 | 15.16 |
| | | | | | | | | 25–35 | 2.91 | |
| | | | | | | | | 36–45 | 3.87 | |
| | | | | | | | | 46–60 | 3.69 | |
| | | | | | | | | >60 | 2.41 | |
| | | | **Total** | | | **486** | **823** | - | - | **44.57** |
| Non family | 1 | 1 | E1 | T21, T17, T18, T40 | H19 | 143 | 241 | - | - | 0 |
| | 2 | 2 | H19 | T11, T29, T26, T66 | H11 | 124 | 210 | - | - | 0 |
| | 3 | 3 | H11 | T25, T24, T15, T22, T57 | E2 | 18 | 31 | - | - | 0 |
| | | | **Total** | | | **285** | **428** | - | - | **0** |

Non-family route planning shows three different routes in family tourism. Because the main factor considered in non-family route planning is the lowest travel cost, the algorithm analyzed popular places with the lowest travel cost. The lowest travel cost of a tourism trip is 428 Bath and zero satisfaction, because satisfaction is not necessary for non-family route planning. The non-family tourism route shows three routes by the number of fixed tourism days. The first day of the tour begins at point E1, which is an airport. The places for sightseeing are T21, T17, T18, and T40, and the first day of the trip ends at H19, which is an accommodation point. The second day starts at point H19, and the sightseeing places are T11, T29, T26, and T66, respectively. The H11 lodging point is the end tour point of the second day. The final tour day starts at point H11 and continues through tourism points T25, T24, T15, T22, and T57.

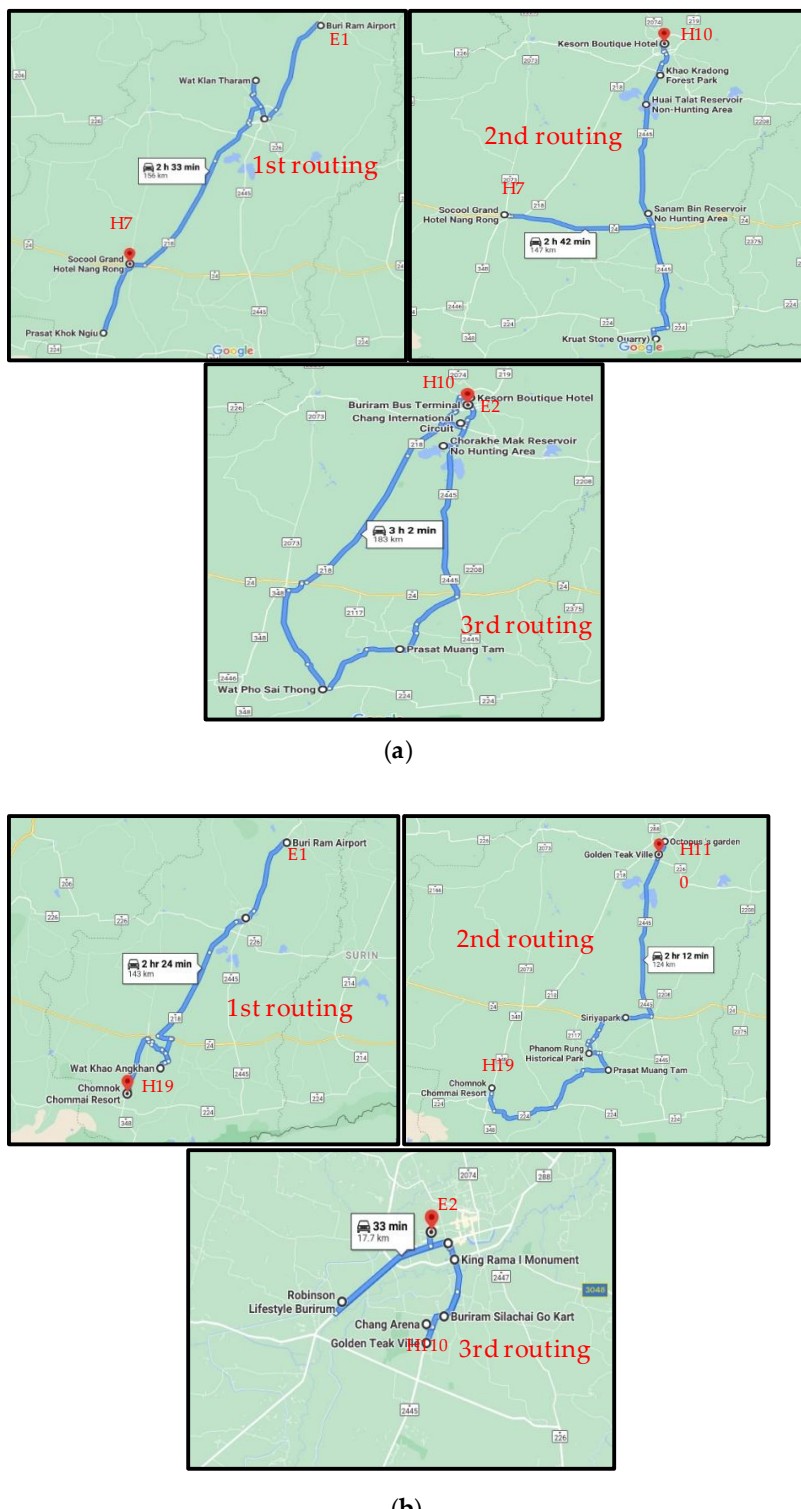

**Figure 11.** Routing case study (**a**) Family tourism routing. and (**b**) Non-family tourism routing.

The tour terminates at point E2 on the final day, which is a bus station. The total travel distance is 285 km, which is the lowest travel distance for non-family tourism route planning. The pattern of the three routes is shown in Figure 11b.

Therefore, the MALNS algorithm shows that it can create an effective route design for family tourism and non-family tourism. This is because the MALNS algorithm can adjust the diversity factor value required by tourists and the optimal route planning generation. In family tourism, route design is particularly difficult due to the differences in terms of the

gender and age ranges. The MALNS algorithm can be used in routing design and optimal routing to maximize family member satisfaction.

In addition, this program can include the selected initial point, tourism place, and endpoint, with the requirement that tourists are flexible and that a fast processing is achieved in tour route planning based on the MALNS algorithm.

## 6. Conclusions and Future Research

This work has considered the family tourism route planning problem. The objective here was to maximize the overall satisfaction rating regarding tourism places with the lowest travel cost. The MALNS algorithm was applied to find optimal solutions for overall satisfaction rating and travel cost, and the tested results were compared with the results for an exact method using the Lingo program. The comparison of the two methods showed similar results for small problem sizes. For medium and large problems, the results of the MALNS algorithm showed a good performance, and a similar performance was found for the Lingo program. The Lingo program showed a greater processing time than the MALNS algorithm. The MALNS algorithm showed a dramatically different processing time. With the MALNS algorithm, the processing time was reduced by 99.94% for medium problem sizes and 99.95% for large problem sizes. The statistical testing of the MALNS algorithm and the Lingo program for all problem sizes showed insignificantly different results in terms of overall satisfaction and travel cost; however, the two methods displayed significantly different processing times for both medium and large problem sizes. Therefore, the MALNS algorithm is an appropriate method for family tourism route design and presents effective solutions with a fast processing time.

The computational result shows that MALNS is effective for finding optimal solutions, meaning that including the four destroy and four repair operators is the best combination. Normally, traditional ALNS includes only one acceptance method to make a decision as to a new solution. However, we design three acceptance criteria. The number of operators and acceptance criteria indicate a high diversification level, which allows the algorithm to escape from the local optima. While it could possibly consume a high processing time, it is well worth the wait. Additionally, we modified the operators for the purpose of intensification, that is, to allow the algorithm to search for the best solution in the local optima. Therefore, the researchers can conclude that the good design of the algorithm plays an important part in the performance and contribution of the MALNS in this research.

In a case study, the MALNS algorithm balanced route planning, while considering the desires of different family members and finding the lowest of traveling cost and distance. The routing proposed in the case study would result in 823 baht for the total traveling cost and a total travel distance of 486 km, which is approximately the minimum traveling cost and distance, with the maximum number of visits to interesting tourist locations, amounting to a travel satisfaction score of 44.57. Therefore, the MALNS method can solve the family tourism route planning problem and provide highly efficient solutions for family members with dissimilar satisfaction rates. The empirical results found for the MALNS algorithm here are very beneficial in terms of software application generation in tourism route planning, which is the problem of finding optimal routes with the maximum overall rating satisfaction of family members and lowest travel cost. Therefore, the MALNS algorithm can be suitable for tour route planning, irrespective of age, gender, and satisfaction differences between family members and the number of tourism places in each route. The balancing of satisfaction and number of tourism places is a target of family tour route planning, and the MALNS algorithm shows high performance in achieving this balance.

However, the MALNS algorithm can be applied to various models of tourism route planning, despite the differences in the characteristics of the problem. When some factors have been added into the model, such as single parents and many family generations and age ranges, we can adjust the model by adding more indices and parameters, then solve using the MALNS in the same way. The aspects of the problem are common to every family. Thus, we believe that the model can be implemented in different parts of the world.

The application of the case study based on the MALNS algorithm allowed for the generation of the software application model and shows an example of travel routing in Figure 12. This software application model is easy to use, flexible, and outcomes are produced very quickly. In addition, this software can identify the requirements of tourists to achieve optimal routing, and it uses less time in the routing process. The application of family tourism planning based on the MALNS algorithm will be used to plan tourism routes for a range of families in support of tourism in Burirum Province, Thailand. The application can provide optimal travel routing with the lowest traveling cost and maximum number of tourism places. It therefore allows tourists to see more sightseeing points, which increases their satisfaction. There is a family tourist influx into Burirum Province. Therefore, the local economy is spending more on tourism, thus increasing the income of the local people and progressing toward a circular economy.

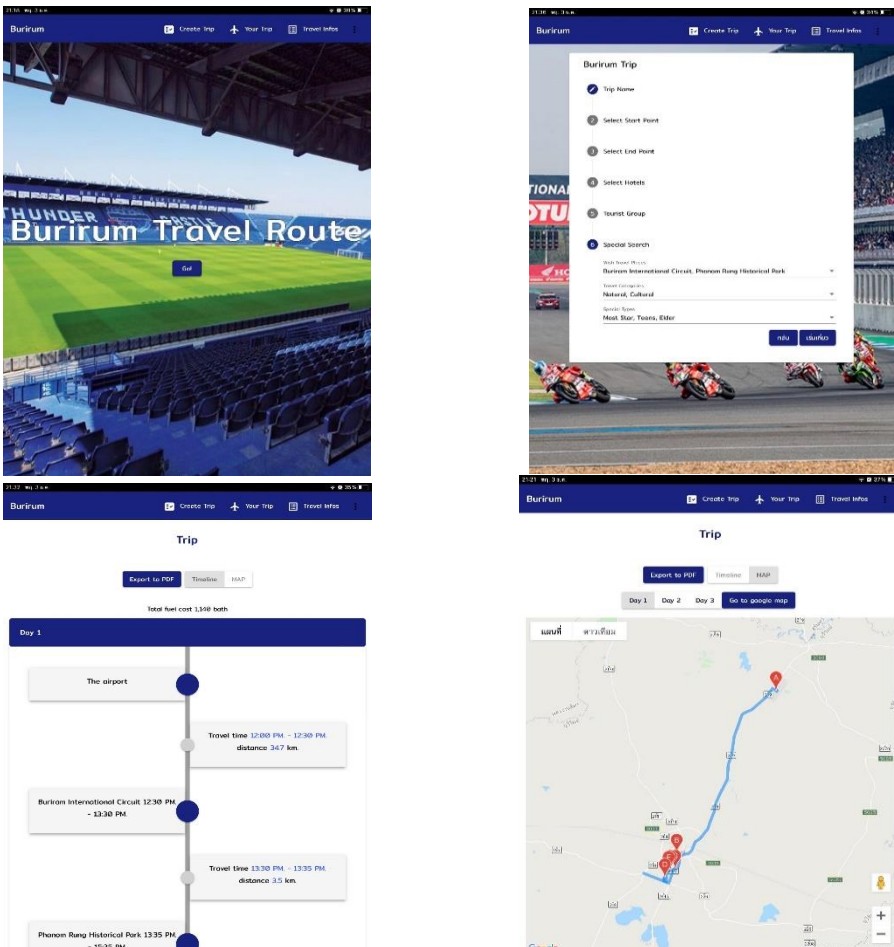

**Figure 12.** Example of software application model.

In the future, the MALNS algorithm will be used in combination with the full function of and tourism data provided by the software development of an application used on mobile phones and tourism webpages in Thailand. The application will support family or group tours, and tourists will be able to easily and independently plan tourism routes.

**Author Contributions:** Conceptualization, N.K. and K.C.; review—editing, N.K. and K.C.; Conceptualization, R.P.; project administration, R.P.; methodology, W.S. and C.T.; review—editing and validation, W.S.; writing—original draft preparation, C.T.; software, C.T.; formal analysis, W.S. and C.T. All authors have read and agreed to the published version of the manuscript.

**Funding:** This research received no external funding.

**Data Availability Statement:** Data is contained within the article.

**Conflicts of Interest:** The authors declare no conflict of interest.

## Appendix A

**Table A1.** Dataset input for testing instances.

| Orders | Tourism Places | Open-Close Times | Latitudes | Longitudes |
|---|---|---|---|---|
| T1 | Dong Yai Wildlife Sanctuary | 08.30–16.30 | 14.240417 | 102.620444 |
| T2 | Sanam Bin Reservoir No Hunting Area | 08.00–17.00 | 14.639476 | 103.069393 |
| T3 | Chorakhe Mak Reservoir No Hunting Area | 08.00–17.00 | 14.913453 | 103.043102 |
| T4 | Huai Talat Reservoir Non-Hunting Area | 08.00–18.00 | 14.887913 | 103.081934 |
| T5 | Lam Nong Rang reservoir | 08.30–22.00 | 14.298851 | 102.757528 |
| T6 | Serrow Ground Walking Street | 16.00–22.00 | 14.996541 | 103.108139 |
| T7 | Buriram castle | 10.00–22.00 | 14.966145 | 103.090289 |
| T8 | Prang Ku Suantang | 08.00–17.00 | 15.554656 | 102.840855 |
| T9 | Prasat Nong Bua Rai | 08.00–17.00 | 14.531549 | 102.962389 |
| T10 | Prasat Ban Bu | 08.00–17.00 | 14.533736 | 102.979201 |
| T11 | Prasat Muang Tam | 08.00–17.00 | 14.496549 | 102.982363 |
| T12 | Prasat Nong Hong | 08.00–17.00 | 14.302663 | 102.760562 |
| T13 | Wat Nong Ya Plong | 08.00–17.00 | 15.056211 | 102.980211 |
| T14 | Wat Rahan | 08.00–17.00 | 15.135111 | 103.190528 |
| T15 | King Rama 1 Memorial | 08.00–17.00 | 14.986743 | 103.104602 |
| T16 | Khao Kradong Forest Park | 08.00–18.00 | 14.939354 | 103.092853 |
| T17 | Wat Klang Phra Aram Luang | 06.00–20.00 | 14.995535 | 103.112385 |
| T18 | Wat Khao Angkhan | 08.00–17.00 | 14.534344 | 102.834489 |
| T19 | Wat Pa Khao Noi | 06.00–18.00 | 14.927719 | 103.094438 |
| T20 | Wat Pho Sai Thong | 08.00–17.00 | 14.413680 | 102.853927 |
| T21 | Buriram City Pillar Shrine | 06.00–17.00 | 14.996274 | 103.113767 |
| T22 | Buri Ram Northeast Culture Center | 08.30–16.30 | 14.991944 | 103.102233 |
| T23 | Chang International Circuit | 09.00–17.00 | 14.958034 | 103.084906 |
| T24 | Buriram Silachai Go Kart | 10.00–19.00 | 14.968361 | 103.100867 |
| T25 | Chang Arena | 09.00–16.00 | 14.965882 | 103.094301 |
| T26 | Siriya park | 09.00–17.00 | 14.608271 | 103.021617 |
| T27 | Shiva Garden 12 | 08.00–22.00 | 14.966788 | 103.090878 |
| T28 | Ban Khok Muang agricultural tourism attraction | 08.30–16.00 | 14.530151 | 103.010062 |
| T29 | Phanom Rung Historical Park | 06.00–18.00 | 14.532078 | 102.942601 |
| T30 | Play La Ploen Flower Park | 09.00–18.00 | 15.283722 | 103.005122 |
| T31 | Wat PA Samakkhi Tham | 08.00–17.00 | 16.272648 | 103.711923 |
| T32 | Wat Ban Sa-Nuan Nok | 08.00–17.00 | 14.937240 | 103.182512 |
| T33 | Rao Su Monument | 08.00–17.00 | 14.299137 | 102.743726 |
| T34 | Ban Kruat Stone Quarry | 08.00–17.00 | 14.364984 | 103.085402 |
| T35 | Play water Park Buriram | 10.00–19.00 | 15.030987 | 103.155851 |

**Table A1.** *Cont.*

| Orders | Tourism Places | Open-Close Times | Latitudes | Longitudes |
|--------|----------------|------------------|-----------|------------|
| T36 | Castle Gold Beach Ki | 06.00–19.00 | 14.713447 | 102.558665 |
| T37 | Naramit water Park Buriram | 10.00–19.00 | 15.039866 | 103.087928 |
| T38 | Thung Laem Reservoir | 08.00–17.00 | 14.632369 | 102.830597 |
| T39 | Wat Chaniang Wanaram | 08.00–17.00 | 14.831796 | 103.220264 |
| T40 | Prasat Khok Ngiu | 08.00–17.00 | 14.463453 | 102.727723 |
| T41 | Ban Suan Fruit Gardens | 08.00–18.00 | 15.160366 | 102.962050 |
| T42 | Prasat Nongkong | 08.00–17.00 | 14.642351 | 102.905577 |
| T43 | Ta Phraya National Park | 08.30–18.00 | 14.129175 | 102.581672 |
| T44 | Red Cliff View Point | 00.00–24.00 | 14.147133 | 102.666129 |
| T45 | Wat Hong | 08.00–17.00 | 15.543270 | 103.016730 |
| T46 | Ban Sa-Nuan Nok | 00.00–24.00 | 15.716630 | 103.173034 |
| T47 | Wat Pho Yoi | 08.00–17.00 | 14.440276 | 102.717500 |
| T48 | Wat Phra Mother Nijjanukhroh Nang Rong | 08.00–17.00 | 14.631288 | 102.785470 |
| T49 | Buriram Monsters Fishing Park | 08.00–18.00 | 15.016772 | 103.001397 |
| T50 | Muang Buri Ram Municipal Night Bazaar | 15.00–23.00 | 14.992911 | 103.108139 |
| T51 | Wat Klan Tharam | 08.00–17.00 | 15.090949 | 103.092611 |
| T52 | Ching Nam Kha Mu Nang Rong | 06.00–20.00 | 14.633679 | 102.795350 |
| T53 | Klim Kitchen | 10.30–14.00 17.00–22.00 | 15.003997 | 103.107161 |
| T54 | White Cottage Cafe | 10.00–20.00 | 14.633885 | 102.787910 |
| T55 | A day awesome Cafe | 07.00–22.00 | 14.994183 | 103.097209 |
| T56 | Cafe de bu | 07.00–23.00 | 14.977417 | 103.106054 |
| T57 | Robinson lifestyle burirum | 10.00–20.00 | 14.973447 | 103.064189 |
| T58 | Ban Mee Steak x Fresh Milk Buriram | 17.30–24.00 | 14.995841 | 103.092913 |
| T59 | Bobby Bang Cafe | 09.30–19.00 | 15.546370 | 102.993728 |
| T60 | Imagine CAFE | 12.00–23.00 | 14.987923 | 103.104390 |
| T61 | Wake Up Cafe' | 08.00–19.00 | 14.997175 | 103.100263 |
| T62 | on the way cafe | 15.00–23.45 | 14.631767 | 102.785780 |
| T63 | Sweet Egg Sushi House | 10.00–20.00 | 14.970729 | 103.133670 |
| T64 | Chumpon Patongko | 04.00–19.00 | 15.002745 | 103.108672 |
| T65 | Jae Noy Saphan Yao | 09.00–16.00 | 14.993677 | 103.111922 |
| T66 | Octopus's garden | 11.00–21.00 | 14.987799 | 103.107456 |
| T67 | Khu Mueang Roast Duck | 07.00–17.00 | 14.958004 | 103.094142 |
| T68 | Standing Meatballs at the train station | 10.00–21.00 | 15.003090 | 103.108253 |
| T69 | Sida Grilled Chicken | 11.00–20.00 | 14.995034 | 103.118918 |
| T70 | Ban shai Nam | 16.00–24.00 | 14.989808 | 103.119605 |
| T71 | Kij Ngam Loed Kha Mu | 08.00–16.00 | 14.984347 | 103.120252 |
| T72 | Lakkhana Nang Rong | 08.00–20.00 | 14.633340 | 102.795280 |
| T73 | Guangzhou buriram | 08.00–15.00 | 15.002965 | 103.106105 |

**Table A1.** *Cont.*

| Orders | Tourism Places | Open-Close Times | Latitudes | Longitudes |
|---|---|---|---|---|
| T74 | Tee Part 2 | 17.00–23.59 | 14.999618 | 103.105532 |
| T75 | Song Pee Nong Restaurant (Dinosaur) Buriram | 08.00–20.00 | 14.917424 | 103.078871 |
| T76 | Gomain boiled pork blood | 05.00–13.00 | 14.994604 | 103.109941 |
| T77 | Baan Lilil Coffee | 10.30–21.00 | 14.969631 | 103.063163 |
| T78 | Phoemphun Kha Moo | 06.00–13.00 | 15.023245 | 102.829763 |
| T79 | Chumpon Coffee | 04.00–09.00 | 15.002739 | 103.108663 |
| T80 | Pa Nee meat ball stand | 10.00–21.00 | 15.003088 | 103.108252 |
| T81 | Muang Pae Shop | 07.00–18.00 | 14.990864 | 103.090475 |
| T82 | Lai Mai Restaurant | 07.00–18.00 | 14.999351 | 103.107723 |
| T83 | Ari Thai Silk | 07.00–18.00 | 14.999348 | 103.107719 |
| T84 | Liang Huat Restaurant | 07.00–18.00 | 14.998902 | 103.110298 |
| T85 | Na Pho District Handicraft Center | 07.00–18.00 | 15.645600 | 102.950203 |
| T86 | Tum & Tum souvenir shop Krayasart | 07.00–18.00 | 14.603205 | 103.078629 |
| T87 | Buriram United Shop | 10.00–21.00 | 14.965568 | 103.095027 |
| T88 | Amari Buriram United | 08.00–24.00 | 14.968069 | 103,093339 |
| H1 | X2 Vibe Buriram Hotel | 08.00–24.00 | 14.973765 | 103.069453 |
| H2 | Best Western Royal Buriram Hotel | 08.00–24.00 | 14.991152 | 103.094472 |
| H3 | Cresco Hotel–Buriram | 08.00–24.00 | 14.971895 | 103.105407 |
| H4 | Furtune Buriram Hotel | 08.00–24.00 | 14.979813 | 103.074035 |
| H5 | T-rex Buriram Boutique Hotel | 08.00–24.00 | 14.933959 | 103.087989 |
| H6 | The Sita Princess Bururam | 08.00–24.00 | 14.995397 | 103.094166 |
| H7 | Socool Grand Hotel | 08.00–24.00 | 14.636402 | 102.790875 |
| H8 | Phanomrungpuri Boutique Hotels & Resorts | 08.00–24.00 | 14.646424 | 102.795033 |
| H9 | Eireann Hotel | 08.00–24.00 | 14.610957 | 103.059917 |
| H10 | Kesorn Boutique | 08.00–24.00 | 15.010318 | 103.101378 |
| H11 | Golden Teak Ville | 08.00–24.00 | 14.959897 | 103.094365 |
| H12 | Green@Buriram Hotel | 08.00–24.00 | 14.988798 | 103.109419 |
| H13 | D Sine Resort | 08.00–24.00 | 14.971569 | 103.086144 |
| H14 | The Zell: Budget Hotel Buriam | 08.00–24.00 | 14.992338 | 103.119848 |
| H15 | Lemon Resort | 08.00–24.00 | 15.020076 | 103.108744 |
| H16 | Rey Mysterio | 08.00–24.00 | 14.999543 | 103.092595 |
| H17 | Klim Hotel | 08.00–24.00 | 15.004273 | 103.106952 |
| H18 | Hotel SG Hotel | 08.00–24.00 | 14.987895 | 103.084897 |
| H19 | Chomnok chommai resort | 08.00–24.00 | 14.457981 | 102.726393 |
| H20 | Modena by Fraser Buriram | 08.00–24.00 | 14.963608 | 103.093613 |
| H21 | Hotel de l'amour Buriram | 08.00–24.00 | 14.610994 | 103.07302 |
| H22 | Play La Ploen | 08.00–24.00 | 15.285704 | 103.002935 |
| H23 | Buriram Judy Park & Resort | 08.00–24.00 | 14.954636 | 103.096126 |
| H24 | Infinity Seesun Resort | 08.00–24.00 | 14.890068 | 103.190117 |

**Table A1.** *Cont.*

| Orders | Tourism Places | Open-Close Times | Latitudes | Longitudes |
|--------|----------------|------------------|-----------|------------|
| H25 | Srianunt Boutique Hotel | 08.00–24.00 | 14.970361 | 103.077076 |
| H26 | Thada Chateau Hotel | 08.00–24.00 | 14.977452 | 103.106681 |
| H27 | Golden Teak Ville | 08.00–24.00 | 14.959972 | 103.094419 |
| H29 | NP Hotel | 08.00–24.00 | 15.009458 | 103.107609 |
| H29 | Pattara House | 08.00–24.00 | 15.010302 | 103.086077 |
| E1 | Airport | Start-End point | 15.228992 | 103.246798 |
| E2 | Bus station | Start-End point | 15.295488 | 103.292935 |
| E3 | Train station | Start-End point | 15.003354 | 103.107981 |

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
