# Peer review of "Modified ALNS Algorithm for a Processing Application of Family Tourist Route Planning: A Case Study of Buriram in Thailand"

_computation, doi:10.3390/computation9020023_

Round 1
Reviewer 1 Report
This paper presents a problem of planning tourist routes or designing tourist routes for a group of tourists, in this case, a family. Itineraries for several days are designed considering categories of interest points and different preferences for each group member.
Tourist route design models have been widely discussed in the literature. The proposed model of the Tour Route Planning Problem (TRPP) has not been approached from familiar routes with categories of preferences over points of interest. The metaheuristics are known but have been modified, adapting it to solve this problem.
The problem is of practical interest, and both the model and the proposed method for finding solutions is correct. The results of the experimentation clearly justify the conclusions.
Some issues need to be improved, which are listed below.
Abstract:
The term time window is a well-studied term in many routing models, almost always associated with points of interest. In my opinion, to say that “solving within various time windows each day”, implying that the model is a multi-window issue that it is not.
It is imprecise and unclear to say that “are approximately 0.71% different than the values derived from the exact method” and also that, “suggesting that the MALNS method could be a great competitive algorithm”.
1-Introduction:
The last contribution or conclusion needs to be better described and justified, concerning the results of the experimentation “Finally, the family tourism application base on the MALNS algorithm is flexible to rerouting and……”
2-Literature Review:
Review the first paragraph and write it better. There are not three modelled problems. POIS is not a problem dealt with in the literature (Remove throughout the manuscript). The complexity of the problems is not only associated with the amount of data. Local search is a heuristic method. The exact methods do not find a solution in reasonable times, and it is necessary to use approximate, heuristic and metaheuristic methods.
The comment to the reference Yu et al. [19] Yu et al.[19]displayed that the team orienteering problem with time windows (TOPTW) is the obstacle to tourist trip design. Is not correct.
In the last paragraph, check: in line 143 the term “variable route design”, in line 148 “statisticians and researchers”, are not exclusive and the phrase “ALNS algorithms generate highly effective solutions and the required processing time is the speed of problem solving alone”.
It is of interest to make clear which are the problems and models treated that are like the one proposed and their differences.
3- Definitions and Formulations Problem
A definition as clear and formal as possible is necessary—a definition of the objective and constraints. We are facing a problem modelled in the literature. Although it helps to describe the problem, giving a real example is not enough.
In the formulation, line 266, the Objective is not a constraint.
In my opinion, the last paragraphs have more to do with experimentation and the definition of instances than with the formulation of the model.
4-Procedure
Do not use Table for Algorithm.
In subsection 4.3 review the wording of the first paragraph, it isn't clear.
Figure 9 is not displayed completely.
5-Experimentation
I have doubts about whether solutions could be found with fewer routes, when 3 or 2 fixed routes are established and why in the experimentation only instances with 3 routes are reached when the description of the instances in section 3 is described in 1-5 days.
Explain the term used in Table 10 "Satisfaction rating per person”.
Figure 10 is not displayed completely.
6.-Conclusions
Explain the term balance used here and, in the introduction, refer to the algorithm.
It is necessary to contribute some line of future research.
Author Response
List of correction of Reviewer 1
1.The term time window is a well-studied term in many routing models, almost always associated with points of interest. In my opinion, to say that “solving within various time windows each day”, implying that the model is a multi-window issue that it is not.
- In this work, the time window isn’t a multi-window. Therefore, we are correction of first sentence in abstract to “This research presents family tourism route problem solving by considering time windows in each day.”
- It is imprecise and unclear to say that “are approximately 0.71% different than the values derived from the exact method” and also that, “suggesting that the MALNS method could be a great competitive algorithm”.
- We describe performance result expansion of abstract on line 25-33.
- The last contribution or conclusion needs to be better described and justified, concerning the results of the experimentation “Finally, the family tourism application base on the MALNS algorithm is flexible to rerouting and……”
- We add more description on line 102-104.
- Review the first paragraph and write it better. There are not three modelled problems. POIS is not a problem dealt with in the literature (Remove throughout the manuscript).
- We remove all the POIs word in the manuscript.
- The complexity of the problems is not only associated with the amount of data. Local search is a heuristic method. The exact methods do not find a solution in reasonable times, and it is necessary to use approximate, heuristic and metaheuristic methods.
- We describe more reason for the problem complexity on line 159-163.
- The comment to the reference Yu et al. [19] “Yu et al.[19]displayed that the team orienteering problem with time windows (TOPTW) is the obstacle to tourist trip design. Is not correct.
- We improve the literature review description of Yu et al. on line 183-190..
- In the last paragraph, check: in line 143 the term “variable route design”,
- We improve the sentence in term “variable route design”on line 205-206. “This is a factor that affects personalized route design in urban tourism and has not been studied in the context of the group tourism problem solving.”
- in line 148 “statisticians and researchers”, are not exclusive and the phrase “ALNS algorithms generate highly effective solutions and the required processing time is the speed of problem solving alone”.
- We correct the sentence on line 227-231.
- A definition as clear and formal as possible is necessary—a definition of the objective and constraints. We are facing a problem modelled in the literature. Although it helps to describe the problem, giving a real example is not enough.
- We give a definition of the objective and constraints in subsection 3.1
- In the formulation, line 266, the Objective is not a constraint.
- We correct “Constraint” to “Objective function” on line 354.
- In my opinion, the last paragraphs have more to do with experimentation and the definition of instances than with the formulation of the model.
- We move the last paragraphs of section 3 to section 5.
- Do not use Table for Algorithm.
- We change the word from “table” to “algorithm” in Section 4.
- In subsection 4.3 review the wording of the first paragraph, it isn't clear.
- We improve words of the first paragraph on line 496-498 of subsection 4.3
- Figure 9 is not displayed completely.
- We completely improve of the figure 9 in page 16.
- I have doubts about whether solutions could be found with fewer routes, when 3 or 2 fixed routes are established and why in the experimentation only instances with 3 routes are reached when the description of the instances in section 3 is described in 1-5 days.
- We describe the routing of case study experiment as follow;
The travel route depends on the fixed number of tourism day such as the tourism day was fixed to 3 days, the result of travel routing is 3 routes. However, this 3 routes are optimal routing on the lowest travel cost. For section 3, the 1-5 tourism days is a range of real dataset of holiday in Thailand. The tourism day can be extended more than the 5 day or the requirement of tourist.
Therefore, this case study determines the 3 tourism day that the travel routing result is 3 routes as shown in table 8
- Explain the term used in Table 10 "Satisfaction rating per person”.
- We change the term “Satisfaction rating per person” to “Satisfaction rating per route” and the table 10 in manuscript changed to the table 8 in page 25.
- Figure 10 is not displayed completely.
- The figure 10 was completely changed and improved to figure 11 in page 24.
- Explain the term balance used here and, in the introduction, refer to the algorithm.
- We described the term “balance” on line 848-856 of conclusion and line 105-108 of introduction.

Reviewer 2 Report
This paper addresses an interesting topic , but it has major problems: 1. When discussing his contribution, the author emphasizes that there is no relevant research that focuses on family tourism route design, and then the relevant research has been published in Tourism Management. Therefore, the author not only did not comprehensively sort out the relevant research, but also did no contribution; 2. The core of the entire research in this paper is the algorithm designing, but the contributions about the algorithm designing is not seen in this research. 3.The author focuses on heuristic algorithms. However, the author does not compare it with existing heuristic algorithms in this research.
Author Response
List of correction of Reviewer 2
- When discussing his contribution, the author emphasizes that there is no relevant research that focuses on family tourism route design, and then the relevant research has been published in Tourism Management. Therefore, the author not only did not comprehensively sort out the relevant research, but also did no contribution;
- We added the relevant research about the family tourism and improved the description of family tourism route design of section 2 in page 3-6.
- The core of the entire research in this paper is the algorithm designing, but the contributions about the algorithm designing is not seen in this research.
- We improved the result description of algorithm design for this work by the comparison of family tourism and non-family tourism in the table 8, in subsection 5.2 on line 733-805; and in section 6 on line 833-842.
3.The author focuses on heuristic algorithms. However, the author does not compare it with existing heuristic algorithms in this research.
- The ALNS algorithm and MALNS algorithm were compared in table 5. The other algorithm performances were compared in Table 6 in subsection 5.1 on line 709-724.

Reviewer 3 Report
Dear Authors,
As a researcher and a mother travelling with kids a lot I like this article very much and I can see a big potential in your idea. As family tourism is one of the most important sectors of the tourism around the world, your subject fits global trends. It is also a crucial part of travel market.
Good trip planning supported by tourism informational system is necessary for positive experiences, visiting more places, and thus - a better effect for the economy. I am sure your methods and research could help in tourism management in Buriram, Thailand. The adaptive large neighbourhood method is mostly used in vehicle and location routing and joint distribution. Your paper convinces that the modified adaptive large neighbourhood search (MALNS) algorithm could be an appropriate method for family tourism planning.
I have some comments though. First of all, in my opinion, abstract should be more clear for those, who are not familiar with this subject. For example, readers might not know what is the exact method or what exactly is small, medium and large scale.
Furthermore, in the text and especially in the Conclusions there is the Lingo program mentioned many times, but it is not mentioned in the Abstract or even in Introduction and it has not been explained clearly enough. Please note that an Abstract should clearly explain: problem, methods, and results.
Moreover, in the Introduction you have mentioned (71-73 line) that tourism is a major target for economic recovery. But you did not mention in the Conclusions how exactly the MALNS could improve economy growth in Buriram.
In my opinion, the conclusions should be rewritten , namely how some results, that is MALNS method can fit for various models of family (single parents, many generations families etc.).
Furthermore, consider explaining in a great details how this model can be implemented in a different parts of the world. This might be a great part of this paper.
Please explain why in Table 9. (line 606) the age range starts from above 25 years, thus you do not consider younger people and children.
The graphical side of article must be improved - some tables and maps are cut in the end of page. Arrows marked on tables blur the text. If possible - entire tables might be shown on one page and not be cut in the middle.
As Buri Ram means the city of happiness - I wish you all a lot of happiness and good luck!

Author Response
List of correction of Reviewer 3
- First of all, in my opinion, abstract should be more clear for those, who are not familiar with this subject. For example, readers might not know what is the exact method or what exactly is small, medium and large scale.
- We wrote the description of abstract including method and type of problem size in Section 5 on line 622-645.
- Furthermore, in the text and especially in the Conclusions there is the Lingo program mentioned many times, but it is not mentioned in the Abstract or even in Introduction and it has not been explained clearly enough.
- We added the relative description in term of lingo program in Abstract part, Introduction part and Computation result part on line 20-21, 108-109 and line 622-624, respectively.
- In the Introduction you have mentioned (71-73 line) that tourism is a major target for economic recovery. But you did not mention in the Conclusions how exactly the MALNS could improve economy growth in Buriram.
- We gave the description of economic recovery in section 6 on line 871-873.
4.In my opinion, the conclusions should be rewritten , namely how some results, that is MALNS method can fit for various models of family (single parents, many generations families etc.). And Furthermore, consider explaining in a great details how this model can be implemented in a different parts of the world. This might be a great part of this paper.
- We rewrite the conclusion and add the description of various models of family and implement in a different parts of the world in section 6 on line 857-862.
- Please explain why in Table 9. (line 606) the age range starts from above 25 years, thus you do not consider younger people and children.
- In table 9, “> 25 ” was changed to “1-24” in table 7.
- The graphical side of article must be improved - some tables and maps are cut in the end of page. Arrows marked on tables blur the text. If possible - entire tables might be shown on one page and not be cut in the middle.
- We improved all graphical side of article completely.

Round 2
Reviewer 1 Report
I agree that most of the issues have been resolved. The manuscript has been corrected and improved
Author Response
Thank you very much for all comment in manuscript improvement.

Reviewer 2 Report
I don't see any modifications to my problem.The author said that they had addedto literature on group tour route design, but in the article they still said that there is no research on group tour route design at present. In addition, the authors say they have done a comparison of the algorithms, but Tables 5 and 6 show the comparison of the authors' algorithms with the software, still not compared with the other algorithms.
Author Response
We corrected the manuscript from comment of reviewer 2. The detail of correction showed as in file attachment.

Reviewer 3 Report
Dear Authors,
Thank you for improving article. I accept it in present form.
Good job and good luck!
Author Response
Thank you very much in all comment for manuscript improvement.

Round 3
Reviewer 2 Report
This version can be accepted.